# Noisy Multi-Label Learning through Co-Occurrence-Aware Diffusion

**Senyu Hou**
School of Computer and Information
Technology, Shanxi University
housenyu@sxu.edu.cn

**Yuru Ren**
School of Computer and Information
Technology, Shanxi University
renyuru@sxu.edu.cn

**Gaoxia Jiang**
School of Computer and Information
Technology, Shanxi University
jianggaoxia@sxu.edu.cn

**Wenjian Wang** [*]
Key Laboratory of Data
Intelligence and Cognitive Computing of
Shanxi Province
wjwang@sxu.edu.cn

## Abstract

Noisy labels often compel models to overfit, especially in multi-label classification tasks. Existing methods for noisy multi-label learning (NML) primarily follow a discriminative paradigm, which relies on noise transition matrix estimation or small-loss strategies to correct noisy labels. However, they remain substantial optimization difficulties compared to noisy single-label learning. In this paper, we propose a **C**o-Occurrence-**A**ware **D**iffusion (CAD) model, which reformulates NML from a generative perspective. We treat features as conditions and multi-labels as diffusion targets, optimizing the diffusion model for multi-label learning with theoretical guarantees. Benefiting from the diffusion model's strength in capturing multi-object semantics and structured label matrix representation, we can effectively learn the posterior mapping from features to true multi-labels. To mitigate the interference of noisy labels in the forward process, we guide generation using pseudo-clean labels reconstructed from the latent neighborhood space, replacing original point-wise estimates with neighborhood-based proxies. In the reverse process, we further incorporate label co-occurrence constraints to enhance the model's awareness of incorrect generation directions, thereby promoting robust optimization. Extensive experiments on both synthetic (Pascal-VOC, MS-COCO) and real-world (NUS-WIDE) noisy datasets demonstrate that our approach outperforms state-of-the-art methods.

## 1 Introduction

Multi-label classification is a specialized subfield of classification tasks where each instance is assigned multiple labels, making it inherently more complex than multi-class classification [1]. Modern multi-label classification methods typically leverage deep neural networks (DNNs) trained iteratively on large-scale, accurately annotated datasets [2–6]. However, collecting expert-labelled data in real-world scenarios is time-consuming and expensive. Crowd-sourcing [7] and model-generated labels [8] are commonly employed to mitigate annotation costs, inevitably introducing noisy labels into datasets. Unfortunately, due to their high capacity, DNNs can fit most training data, regardless of whether the labels are clean or noisy [9–12]. This overfitting to noisy labels prevents

---

[*]Corresponding author: Wenjian Wang

39th Conference on Neural Information Processing Systems (NeurIPS 2025).

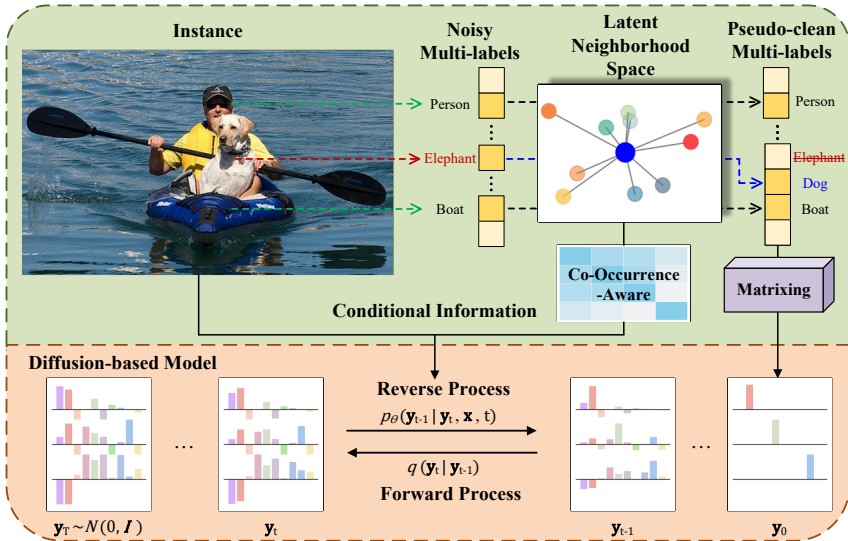

Figure 1: The illustration of the co-occurrence-aware diffusion model. The example comes from the MS-COCO dataset, where the label set has been corrupted to {*Person*, *Elephant*, *Boat*}. The true co-occurrence probability of the label pair $< Elephant, Boat >$ is very low, so our intuition is to constrain the learning weight of this pair during the reverse process. The pseudo-clean label set {*Person*, *Dog*, *Boat*}, estimated through latent neighborhood space, is more suitable as the starting point $\mathbf{y}_0$ for label diffusion after being transformed into matrix form.

the model from achieving actual empirical risk minimization, making it crucial to address label noise in multi-label classification.

To tackle this challenge, researchers have proposed noisy multi-label learning (NML), a paradigm that extends the well-established label noise learning (LNL) from single-label settings to multi-label scenarios [13–17]. Existing NML approaches predominantly follow LNL strategies, employing noise transition matrix estimation [18–21] and correction mechanisms [22–26] to identify and rectify noisy labels. Some studies [27, 20] further exploit label dependencies to improve the accuracy of noise estimation or correction. However, most of these methods are based on discriminative paradigms, often facing optimization challenges.

Huang et al. [28] proposed a generative-based method LSNPC, where they used variational autoencoders (VAE) [29] to treat noisy labels as the result of latent space transitions, enabling more accurate label correction. Similarly, in the LNL domain, Bae et al. [30] utilized VAE to estimate the transition matrix for post-processing label correction, while Chen et al. [31] framed label noise as a random generation process and used denoising diffusion probabilistic models (DDPM) [32] to model the uncertainty in noisy label generation, optimizing classification via maximum likelihood estimation. These works demonstrate the feasibility and effectiveness of generative models in classification tasks. Inspired by this insights, we reformulate the NML problem using DDPM as a robust label generation task.

We aim to develop a universal denoising probabilistic diffusion framework for NML, achieved by modeling multi-labels as diffusion targets and using instance features as guided conditions. By representing the labels in matrix form, we strengthen the mapping between features and multi-labels, ultimately optimizing maximum likelihood estimation to obtain the accurate class distribution. A key challenge in this paradigm is that only noisy multi-labels are available. To overcome this issue, we reconstruct pseudo-clean labels based on each sample's latent neighborhood feature space, and estimate a meta-label co-occurrence matrix from a relatively clean subset to guide the reverse process. Figure 1 illustrates our intuition: when noisy labels exhibit low co-occurrence probability (e.g., *Elephant* vs. *Boat*), our model learns to steer generation away from such incorrect associations via co-occurrence constraints. The latent neighborhood feature space is extracted using a pre-trained encoder, obtained through self-supervised learning or large-scale pre-trained models, enhancing robustness against noise. These pre-trained encoders remain frozen during diffusion model training, ensuring flexibility and broad applicability.

Our main contributions are summarized as follows: 1) We reframe NML as a robust label generation task based on diffusion models with theoretical deduction, and enhance the mapping between features and multi-labels through matrix-based label representation. 2) We design a pseudo-clean label reconstructor and a meta-label co-occurrence matrix estimator, leveraging pre-trained encoders to provide strong priors for diffusion model training. 3) We integrate co-occurrence constraints into the diffusion modeling, proposing the **C**o-Occurrence-**A**ware **D**iffusion (CAD) model, which can robustly learn the generative mapping from features to true multi-labels. 4) Our model achieves state-of-the-art (SOTA) performance in synthetic and real-world noisy datasets, e.g., 5∼8% OF1 improvement on noisy Pascal-VOC datasets.

## 2 Related Work

**Noisy Multi-label Learning.** Previous studies have primarily focused on improving multi-label classification models. BCE [33], ADDGCN [34] and ASL [35] are highly effective for multi-label classification but lack mechanisms to handle label noise. Numerous LNL approaches, such as GCE [36], Co-teaching [37] and DivideMix [38], aim to develop noise-robust classification models but lack specialized techniques for handling noisy multi-labels. Liu et al. [39] highlighted the lack and urgency of research in NML, and only recently have a few studies emerged to address this issue directly. Hu et al. [40] proposes WSIC using clean subsets to regularize network learning, while Xie et al. [41] introduces a unified learning framework called CCMN for learning with class-conditional noise. Xia et al. [27] explored the critical role of label dependencies in noisy label learning and proposed the holistic label correction (HLC) algorithm to refine multi-label classification under label noise. Our method focuses explicitly on tackling the NML problem and advocates for uising generative models. We model multi-label noise as a stochastic generation process, capturing the uncertainty in multi-label generation to enhance robustness against noise.

**Generative Classification Method.** Bae et al. [30] introduced VAE to estimate the transition matrix for post-processing correction in single-label tasks. At the same time, Huang et al. [28] integrated the VAE with previous NML methods for more accurate label correction. However, their fundamental framework remains limited to a discriminative paradigm. Diffusion models [32, 42–44] were initially designed for image generation tasks, and classification and regression diffusion models (CARD) [45] first treated classification tasks as conditional generative tasks, generating single-label outputs based on features. Building on this, label-retrieval-augmented diffusion models (LRAD) [31] addressed the robustness issue of diffusion models in noisy single-label learning by incorporating a neighbor retrieval mechanism. Our approach redefines NML as robust generative inference learning and enhances the adaptability of diffusion models in transitioning from single-label to multi-label classification tasks by matrixing labels. We also introduce a unique co-occurrence constraint to alleviate the multi-label noise problem further.

## 3 Preliminary

This section introduces the diffusion theoretical framework and training architecture for single-label classification tasks. Similar to traditional DDPM, it consists of a forward diffusion process and a reverse denoising process. The target of diffusion and denoising is the sample labels, with instance features embedded as control conditions in the diffusion paradigm. For example, CARD introduces a pre-trained encoder $f_\phi$. In the forward process, it uses the encoded result of the sample features $f_\phi(\mathbf{x})$ as the Gaussian noise mean, performing conditional diffusion on the one-hot labels with the diffusion endpoint prior distribution as $q(\mathbf{y}_T \mid \mathbf{x}) = \mathcal{N}(\mathbf{y}_T; f_\phi(\mathbf{x}), \mathbf{I})$. Using a time diffusion schedule $\{\beta_t\}_{t=1:T} \in (0,1)^T$, the conditional distribution of the forward process is defined as $q(\mathbf{y}_t|\mathbf{y}_{t-1}, f_\phi(\mathbf{x})) = \mathcal{N}(\mathbf{y}_t; \sqrt{1-\beta_t}\mathbf{y}_{t-1} + (1 - \sqrt{1-\beta_t})f_\phi(\mathbf{x}), \beta_t\mathbf{I})$. This admits a closed-form sampling distribution with an arbitrary time step $t$: $q(\mathbf{y}_t \mid \mathbf{y}_0, f_\phi(\mathbf{x})) = \mathcal{N}(\mathbf{y}_t; \sqrt{\bar{\alpha}_t}\mathbf{y}_0 + (1 - \sqrt{\bar{\alpha}_t})f_\phi(\mathbf{x}), (1 - \bar{\alpha}_t)\mathbf{I})$, where $\alpha_t := 1 - \beta_t$ and $\bar{\alpha}_t := \prod_t \alpha_t$. Following DDPM's approach, the posterior distribution of the forward process can be derived using Bayes' rule:

$$q(\mathbf{y}_{t-1} \mid \mathbf{y}_t, \mathbf{y}_0, \mathbf{x}) := q(\mathbf{y}_{t-1} \mid \mathbf{y}_t, \mathbf{y}_0, f_\phi(\mathbf{x})) = \mathcal{N}\left(\mathbf{y}_{t-1}; \tilde{\mu}_t(\mathbf{y}_t, \mathbf{y}_0, f_\phi(\mathbf{x})), \tilde{\beta}_t\mathbf{I}\right), \quad (1)$$

where $\tilde{\mu}_t = \frac{\beta_t\sqrt{\alpha_{t-1}}}{1-\alpha_t}\mathbf{y}_0 + \frac{(1-\bar{\alpha}_{t-1})\sqrt{\alpha_t}}{1-\bar{\alpha}_t}\mathbf{y}_t + (1 + \frac{(\sqrt{\bar{\alpha}_t}-1)\sqrt{\alpha_t}+\sqrt{\bar{\alpha}_{t-1}}}{1-\bar{\alpha}_t})f_\phi(\mathbf{x})$ and $\tilde{\beta}_t = \frac{1-\bar{\alpha}_{t-1}}{1-\bar{\alpha}_t}\beta_t$.

In the reverse process, CARD gradually recovers the label vector $\mathbf{y}_0$ from a Gaussian noise distribution $p(\mathbf{y}_T \mid \mathbf{x}) = \mathcal{N}(\mathbf{y}_T; f_\phi(\mathbf{x}), \mathbf{I})$ :

$$p_\theta \left(\mathbf{y}_{t-1} \mid \mathbf{y}_t, \mathbf{x}, f_\phi\left(\mathbf{x}\right)\right) := \mathcal{N} \left(\mathbf{y}_{t-1}; \mu_\theta, \tilde{\beta}_t \mathbf{I}\right). \tag{2}$$

The primary objective of model learning is to make the predicted posterior distribution $p_\theta$ approximate the true noise-added estimated posterior distribution $q$. According to the evidence lower bound (ELBO) and Kullback-Leibler (KL) divergence, the optimization objective can be written as:

$$\mathcal{L}_{t-1} := \mathbb{E}_q \left[D_{KL} \left(q \left(\mathbf{y}_{t-1} \mid \left(\mathbf{y}_t, \mathbf{y}_0, f_\phi(\mathbf{x})\right)\right) \| p_\theta \left(\mathbf{y}_{t-1} \mid \left(\mathbf{y}_t, \mathbf{x}, f_\phi\left(\mathbf{x}\right)\right)\right)\right)\right]. \tag{3}$$

Observing that learnable knowledge resides on the distribution's mean term, which can be reparameterized as $\mu_\theta = \frac{1}{\sqrt{\alpha_t}}(\mathbf{y}_t - \frac{\beta_t}{\sqrt{1-\bar{\alpha}_t}}\epsilon_\theta(\mathbf{y}_t, \mathbf{x}, f_\phi\left(\mathbf{x}\right), t))$. The final loss function simplifies to $\mathcal{L}_{simple} = \|\epsilon - \epsilon_\theta \left(\mathbf{y}_t, \mathbf{x}, f_\phi(\mathbf{x}), t\right)\|^2$, which represents the mean squared error between the model-predicted noise $\epsilon_\theta$ and the real random noise $\epsilon$.

## 4    Co-Occurrence-Aware Diffusion Model

In this section, we first provide an overview of our proposed method in Section 4.1, followed by an introduction to the theoretical framework and inference mechanism of conditional multi-label diffusion in Section 4.2. Finally, in Section 4.3, we propose replacing point estimation with neighborhood distribution estimation and introducing multi-label co-occurrence-aware in the reverse process.

### 4.1    Model Overview

Our co-occurrence-aware diffusion (CAD) model addresses the NML problem, with the overall framework shown in Figure 1. We use a diffusion paradigm to model the multi-label matrix, recovering clean labels from a noisy label distribution based on features and co-occurrence awareness. Instead of using point estimates, we replace them with the multi-label distribution of neighboring instances and leverage its stability to extract a subset of metadata (i.e., a clean subset). This strategy enables the estimation of a meta-co-occurrence probability matrix. During the training phase of the diffusion-based model, the estimated neighborhood label distribution is formatted into a matrix structure to serve as the generation target, $\mathbf{y}_0$, which is progressively corrupted into a standard Gaussian distribution, $\mathcal{N}(0, \mathbf{I})$ in the forward process. In the reverse process, we not only utilize feature-conditioned guidance but also incorporate co-occurrence-aware strategy to prevent the generation of unrealistic label combinations.

### 4.2    Diffusion-based Multi-label Learning

Inspired by CARD, we redefine NML as a stochastic process of conditional label generation (i.e., label diffusion). However, there are two issues can be optimized. First, multi-labels inherently offer greater scalability than single-labels, as a multi-label can be extended as $C$ one-hot vectors (denoting C as the number of classes). Therefore, we model the label matrix instead of a single-dimensional label vector. Second, when $f_\phi$ fails to accurately map to the true classes, especially in datasets with label noise, it significantly interferes with the learning direction. As a result, we discard $f_\phi$ and remodel the conditional multi-label diffusion paradigm.

**Forward process.** In the forward process, we redefine a conditional Markov diffusion process similar to the unconditional multi-label forward distribution $q$ (detailed theoretical framework in Appendix A), assuming that when conditioned on features $\mathbf{x}$, the prior distribution of $\hat{q}$ is the same as $q$:

$$\begin{aligned} \hat{q}\left(\mathbf{y}_t | \mathbf{y}_{t-1}, \mathbf{x}\right) &:= q\left(\mathbf{y}_t | \mathbf{y}_{t-1}\right) \\ &= \mathcal{N}\left(\mathbf{y}_t; \sqrt{1-\beta_t}\mathbf{y}_{t-1}, \beta_t \mathbf{I}\right), \end{aligned} \tag{4}$$

$$\begin{aligned} \hat{q}\left(\mathbf{y}_{1:T} \mid \mathbf{y}_0, \mathbf{x}\right) &:= \prod_{t=1}^{T} \hat{q}\left(\mathbf{y}_t \mid \mathbf{y}_{t-1}, \mathbf{x}\right) \\ &= \mathcal{N}\left(\mathbf{y}_t; \sqrt{\bar{\alpha}_t}\mathbf{y}_0, \left(1-\bar{\alpha}_t\right)\mathbf{I}\right), \end{aligned} \tag{5}$$

where parameters $\bar{\alpha}_t$ and $\beta_t$ are consistent with those in DDPM, meaning the forward diffusion process of the labels is not influenced by any features, and the endpoint conditional distribution of label diffusion is $q(\mathbf{y}_T \mid \mathbf{x}) = \mathcal{N}(\mathbf{y}_T; 0, \mathbf{I})$. It is important to note that, unlike single-label classification tasks where the diffusion target is a one-hot vector, our diffusion target is a $C \times C$ one-hot matrix obtained by forming the multi-label into a matrix structure. The higher-dimensional diffusion target is more stable in generative learning compared to a single vector, facilitating the establishment of a stronger mapping from features to true multi-labels.

We derive the conditional posterior distribution during the forward diffusion process using Bayes' rule and Taylor expansion:

$$\hat{q}(\mathbf{y}_{t-1} \mid \mathbf{y}_t, x) = \mathcal{N}(\mathbf{y}_{t-1}; \hat{\mu}_t, \sigma_t^2 \mathbf{I}), \tag{6}$$

where $\hat{\mu}_t = \frac{\beta_t \sqrt{\bar{\alpha}_{t-1}}}{1 - \bar{\alpha}_t} \mathbf{y}_0 + \frac{(1 - \bar{\alpha}_{t-1})\sqrt{\bar{\alpha}_t}}{1 - \bar{\alpha}_t} \mathbf{y}_t + \sigma_t^2 \nabla_{y_{t-1}} \log p(\mathbf{x}|\mathbf{y}_{t-1})$ and $\sigma_t = \sqrt{\frac{1 - \bar{\alpha}_{t-1}}{1 - \bar{\alpha}_t} \beta_t}$. The last term in $\hat{\mu}_t$ can be considered a deviation to the unconditional distribution mean, guided by the first derivative of $p(\mathbf{x}|\mathbf{y}_{t-1})$, indicating that during training, the decoding information from $\mathbf{y}_t$ to $\mathbf{x}$ must be introduced, otherwise the predictions of the diffusion model will become completely random, and no generative knowledge will be learned (See Appendix B for detailed proof).

**Reverse process.** During the reverse generation process, we directly use feature information as input for the neural network model to predict the reverse posterior distribution:

$$p_\theta(\mathbf{y}_{t-1} \mid \mathbf{y}_t, \mathbf{x}, t) := \mathcal{N}\left(\mathbf{y}_{t-1}; \mu_\theta(\mathbf{y}_t, \mathbf{x}, t), \tilde{\beta}_t \mathbf{I}\right). \tag{7}$$

where $\tilde{\beta}_t = \frac{1 - \bar{\alpha}_{t-1}}{1 - \bar{\alpha}_t} \beta_t$. Based on the analysis of the forward process, reverse prediction requires training a class decoder $p(\mathbf{x}|\mathbf{y}_t)$ for gradient guidance, which may incur additional costs. We adopt a straightforward approach which naturally integrates the decoding process into the diffusion model architecture. Specifically, we use a trainable decoding layer $L_1$ to map $\mathbf{y}_t$ into the feature space of $\mathbf{x}$. However, this is impractical due to the much lower dimensionality of $\mathbf{y}_t$ compared to the conditional feature space dimensions. Therefore, we utilize a pre-trained encoder to obtain quality information during the label purification process, concatenating it with the labels for joint decoding. This enriches the original label information, effectively improving the learning quality of the decoding layer. Detailed information about the specific diffusion model network structure is shown in Appendix Figure B.1. By reparameterizing the training objective, our final training loss is simplified to

$$\mathcal{L}_\epsilon = || \epsilon - \epsilon_\theta \left( \sqrt{\bar{\alpha}_t} \mathbf{y}_0 + \sqrt{1 - \bar{\alpha}_t} \epsilon, \mathbf{x}, t \right) ||^2, \tag{8}$$

**Inference.** Since the classification diffusion model is deterministic in inference and the dimensionality of the multi-label vector is much lower than that of features, we mimic the inference process of denoising diffusion implicit models (DDIM) [46] to generate multi-labels in as few steps as possible without affecting the generated results. Appendix C introduces the inference algorithm, its implementation details, and the recorded inference times. In experiments, we use settings of $S = 10$ and $T = 1000$, where the inference efficiency of the diffusion model is on the same order of magnitude as that of traditional classification models (See Appendix Table H.1).

### 4.3 Neighborhood Label Estimation and Co-Occurrence-Aware

By optimizing the diffusion-based multi-label learning, we can effectively guide the model to learn a generative mapping from features to true multi-labels. However, in noisy environments, excessive contamination of the generative target can lead to a wrong shift in learning target distribution. To mitigate this issue, we propose optimizations from both the forward and reverse process simultaneously.

**Neighborhood label estimation for forward process.** A key challenge in label estimation in noisy environments is the reliance on single-point estimates, which may fail to capture wrong label distributions. To address this, we adopt a neighborhood-based label distribution estimation as a proxy for each instance. This approach is grounded in the neighborhood consistency assumption [47], which posits that in a latent space, instances with similar features tend to cluster. Ensuring this assumption holds requires an unbiased latent space, which we achieve by leveraging pre-trained encoders trained via self-supervised learning or on meta-data, thus mitigating the influence of label noise. In fact, pre-training not only significantly enhances adversarial robustness but is also widely used in LNL to improve resilience against noisy labels.

Table 1: The mean and standard deviation of results (%) on noisy Pascal-VOC 2007

| Metrics | Methods | Sym. 10% | Sym. 30% | Sym. 50% | Asym. 10% | Asym. 30% | Asym. 50% |
|---|---|---|---|---|---|---|---|
| mAP ↑ | Standard | 70.17±0.84 | 64.50±1.20 | 48.19±0.23 | 72.66±1.15 | 60.94±4.25 | 46.72±2.13 |
| | ASL | 72.98±0.85 | 66.79±1.19 | 47.64±1.81 | 73.50±0.99 | 61.99±0.78 | 45.10±0.15 |
| | LRAD-R | 74.71±0.61 | 69.97±1.36 | 67.44±0.61 | 74.55±0.79 | 70.64±0.93 | 59.27±0.72 |
| | HLC | 76.59±1.76 | 72.07±0.67 | 68.03±0.78 | 75.72±0.64 | 69.86±1.61 | 59.09±1.73 |
| | LSNPC-R | 79.21±1.16 | 73.27±0.87 | 70.19±0.93 | 77.52±1.53 | 71.33±0.95 | 58.91±0.26 |
| | CAD-R$^\dagger$ | 80.56±0.69 | 75.18±0.47 | 72.37±0.29 | 78.98±0.90 | 74.56±0.45 | 61.88±1.05 |
| | LRAD-V | 82.50±0.24 | 80.14±1.00 | 78.93±0.74 | 80.14±0.99 | 79.94±1.79 | 68.72±1.05 |
| | LSNPC-V | 83.71±0.58 | 80.19±0.80 | 79.69±0.66 | 79.90±0.88 | 80.01±0.55 | 68.57±1.19 |
| | CAD-V$^\dagger$ | 87.82±0.18 | 86.86±0.60 | 84.61±0.58 | 87.50±0.58 | 81.24±0.42 | 69.37±0.41 |
| OF1 ↑ | Standard | 65.51±1.23 | 63.52±0.48 | 48.08±1.77 | 67.34±0.78 | 58.30±2.82 | 42.19±1.11 |
| | ASL | 68.14±0.60 | 65.11±0.77 | 47.65±1.32 | 68.12±1.93 | 60.02±1.80 | 43.83±1.10 |
| | LRAD-R | 69.75±1.11 | 60.88±1.10 | 53.30±0.33 | 69.10±0.96 | 61.42±0.47 | 45.32±1.50 |
| | HLC | 71.51±1.81 | 68.71±0.28 | 66.62±1.52 | 70.18±1.67 | 64.75±0.70 | 58.58±1.35 |
| | LSNPC-R | 73.96±0.50 | 67.75±0.66 | 65.47±1.19 | 71.87±1.15 | 63.08±0.66 | 55.07±0.59 |
| | CAD-R$^\dagger$ | 75.22±0.74 | 71.42±0.92 | 67.19±1.50 | 73.24±0.91 | 65.77±1.72 | 59.51±0.43 |
| | LRAD-V | 76.03±1.83 | 69.47±1.18 | 68.38±1.12 | 74.28±0.72 | 65.37±1.81 | 58.45±1.15 |
| | LSNPC-V | 77.15±1.60 | 70.78±0.19 | 69.08±0.29 | 74.05±1.19 | 66.43±1.36 | 59.35±1.65 |
| | CAD-V$^\dagger$ | 82.00±0.10 | 76.42±0.11 | 72.41±0.62 | 81.40±0.99 | 70.12±0.85 | 64.51±0.46 |
| CF1 ↑ | Standard | 65.65±1.52 | 55.64±0.84 | 43.04±1.71 | 67.16±1.03 | 57.03±3.43 | 41.24±1.75 |
| | ASL | 68.28±0.68 | 57.63±1.27 | 47.60±0.13 | 67.94±0.74 | 58.33±0.49 | 42.84±1.16 |
| | LRAD-R | 69.90±0.87 | 60.37±0.53 | 53.24±1.81 | 68.91±1.80 | 62.05±2.98 | 44.29±0.62 |
| | HLC | 71.66±1.45 | 62.19±0.42 | 63.65±0.36 | 70.07±1.37 | 65.63±1.04 | 56.51±1.94 |
| | LSNPC-R | 74.11±1.71 | 63.22±0.62 | 62.41±0.26 | 71.67±1.74 | 61.49±1.09 | 54.05±0.57 |
| | CAD-R$^\dagger$ | 75.37±0.57 | 64.87±0.44 | 64.13±1.08 | 73.00±0.81 | 65.78±0.52 | 56.17±0.49 |
| | LRAD-V | 77.19±1.53 | 70.87±0.14 | 64.31±0.11 | 74.08±1.19 | 64.67±1.89 | 59.28±1.69 |
| | LSNPC-V | 78.31±1.27 | 69.19±0.76 | 64.92±0.55 | 73.85±1.25 | 64.73±1.38 | 59.18±1.32 |
| | CAD-V$^\dagger$ | 82.17±0.83 | 75.78±0.19 | 69.33±0.72 | 81.18±0.25 | 70.25±0.69 | 62.74±0.54 |

Specifically, given a noisy training dataset $\mathcal{D} = \big\{(x_i, \tilde{y}_i) \mid x_i \in \mathbb{R}^d, \tilde{y}_i \in \mathbb{R}^m, i = 1, \ldots, n\big\}$, for each instance $x_i$ in the latent space, we assign the label sets of its $K$ nearest neighbors, i.e., $\mathcal{Y}_i = \{y_i^{(1)}, \ldots, y_i^{(K)}\}$. Then we compute the normalized frequency distribution of these labels by $\bar{y}_i = \frac{1}{|\mathcal{Y}_i|} \sum_{k=1}^{K} y^{(k)}$, which estimates distribution then serves as the generation target $\mathbf{y}_0$ for diffusion model's training. To this end, the model mimics the human annotation process, where labels are assigned based on contextual information from similar features. This approach parallels how humans retrieve similar features from memory for semantic labeling, allowing the model to learn and infer semantic similarities. While the neighborhood proxy method mitigates errors in the generation target, severe noise contamination can still introduce estimation bias (see detailed analysis in Appendix F). To address this, we incorporate co-occurrence awareness into the reverse process to refine the generation direction.

**Co-occurrence-aware in reverse process.** Our goal is to pre-estimate a label co-occurrence probability matrix that captures relationships between label pairs, guiding the diffusion model to to be aware of implausible label combinations and avoid generating them(e.g., *elephant* vs. *boat*). Ideally, such estimation would rely on a meta-dataset, as seen in meta-learning, to ensure robustness. However, in real-world noisy multi-label learning, directly available meta-data is often lacking. To overcome this, we secondary use the latent neighborhood space to extract a clean meta-subset from noisy data. Specifically, for each instance $x_i$, we estimate its neighborhood label distribution $\bar{y}_i$ while measure its instability by $\delta_i = \frac{1}{|\mathcal{Y}_i|} \sum_{y_j \in \mathcal{Y}_i} (y_j - \bar{y}_i)^2$. To differentiate samples with varying instability, we employ a binary Gaussian mixture model (GMM) to model the distribution of $\delta_i$, and define the meta-subset(clean subset) $\mathcal{D}'$ as the subset of samples belonging to the lower mean component in GMM (i.e., lower instability samples). This aligns with our neighborhood consistency assumption, as samples with more stable neighborhood distributions are more likely to be clean and thus more valuable for estimating the co-occurrence matrix.

For the selected meta-subset $\mathcal{D}'$, we estimate the co-occurrence probability of each label category using the co-occurrence matrix $\mathcal{C}$, where $\mathcal{C}_{m,n} = P(y_m = 1, y_n = 1 | x \in \mathcal{D}')$. Each element $\mathcal{C}_{m,n}$ represents the probability of labels $m$ and $n$ appearing together within the meta-subset. Then, we com-

**Algorithm 1** CAD Training

---

**Input:** noisy training set $\mathcal{D} = \{\mathbf{X}, \tilde{\mathbf{Y}}\}$, pre-trained encoder $f_p$

1: Obtain latent feature space by $f_p$ and record the $K$-nearest neighbor of each sample
2: Compute instability scores $\delta$ and isolate meta-subset $\mathcal{D}'$
3: Estimate co-occurrence matrix $\mathcal{C}$ on $\mathcal{D}'$
4: **while** not converged **do**
5:   Sample a batch data $(\mathbf{x}, \tilde{\mathbf{y}}) \sim \mathcal{D}$; time slice $t \sim \{1, \ldots, T\}$; and noise $\epsilon \sim \mathcal{N}(0, \mathbf{I})$
6:   Estimate $\bar{\mathbf{y}}$ as $\mathbf{y}_0$ and convert it to a one-hot matrix
7:   Compute learning weight $\mathbf{w}$ and take gradient descent step on the loss $\mathcal{L}'_\epsilon$ (Eq.9)
8: **end while**

---

pute the mean co-occurrence rate across all label pairs for samples $x_i$ by $\bar{\mathcal{C}}_i = \frac{1}{|\mathcal{Y}_i|} \sum_{(y_m, y_n) \in \mathcal{Y}_i} \mathcal{C}_{m,n}$. Note that we use a hard truncation to count co-occurrence label combinations $\mathcal{Y}_i$, i.e., a positive label is assigned if its value is greater than 0.5. Finally, we apply Min-Max normalization at the batch level to obtain the learning weight $w_i$ for each sample, which is used to adjust the learning weight by modifying Eq. 8:

$$\mathcal{L}'_\epsilon = \mathbf{w} \cdot || \epsilon - \epsilon_\theta \left( \sqrt{\bar{\alpha}_t} \bar{\mathbf{y}} + \sqrt{1 - \bar{\alpha}_t} \epsilon, \mathbf{x}, t \right) ||^2, \tag{9}$$

Algorithm 1 describes the overall training procedure. Steps 1–3 correspond to data preparation, while steps 4–8 cover diffusion model training. After obtaining the trained diffusion model and its parameters, inference is performed using Appendix Algorithm C.1, which progressively generates the label matrix from input instances. Finally, the multi-label set is recovered through matrix inversion.

## 5 Experimental Results and Analysis

### 5.1 Experimental Setup

**Datasets and noisy multi-label simulation.** We validate the effectiveness of the proposed method on three multi-label synthetic noisy datasets: Pascal-VOC 2007 [48], Pascal-VOC 2012 [48] and MS-COCO [49], as well as on the real-world noisy dataset NUS-WIDE [50]. In the synthetic noisy datasets, we randomly retain 10% of the samples as a validation set and introduce simulated multi-label noise into the training set using a noise transition matrix $T$ [18–21]. Specifically, for any $i \neq j$, $T_{ij} = r \left( y_j \in \tilde{\mathcal{Y}} \wedge y_i \notin \tilde{\mathcal{Y}} | y_j \notin \mathcal{Y} \wedge y_i \in \mathcal{Y} \right)$ represents the probability $r$ of the $i$-th class label to be corrupted into the $j$-th class label. To mimic real-world label noise, we consider both symmetric and asymmetric noise patterns and select noise rates of 10%, 30%, and 50% for each patterns. The details of the transition matrix are provided in Appendix Figure D.1.

**Baselines.** We exploit the following baselines: (1) Standard [33], which trains ResNet with a BCE loss. (2) ASL [35], which operates differently on positive and negative samples. (3) LRAD-R [31], a diffusion model with a pre-trained ResNet encoder. (4) HLC [27], a label correction method using label dependence. (5) LSNPC-R [28], which integrates a VAE with a pre-trained ResNet encoder. (6) LRAD-V [31], which builds upon LRAD-R by replacing the ResNet with a Vision Transformer (ViT) [51] pre-trained encoder for improved feature extraction. (7) LSNPC-V [28], a variation of LSNPC-R, replacing the ResNet with a ViT pre-trained encoder to combine generative modeling with transformer-based representations. Note that Standard, ASL are designed for clean multi-label data. LRAD-R/V is designed for LNL. HLC and LSNPC-R/V are designed for NML.

**Implementation details and metrics.** Our model requires a pre-trained encoder $f_p$. We use two versions, **CAD-R** and **CAD-V**, which integrate ResNet-50 and ViT-14/L models pre-trained on ImageNet, respectively. Both remain frozen during the diffusion model training without fine-tuning. For fairness, baselines (1)–(5) are compared with CAD-R, which called the *Common group*, while baselines (6) and (7) are compared with CAD-V, which called the *ViT group*. We report the mean and standard deviation of the results from five random experiments. Following the conventional settings [52], we evaluate multi-label classification performance using mean average precision (mAP), overall F1 score (OF1), and per-class F1 score (CF1) as assessment metrics. Notably, the best results in *Common group* and *ViT group* are highlighted in blue and red, respectively. More details and pre-trained model combinations' results can be found in Appendix D.2 and Appendix E.

Table 2: The mean and standard deviation of results (%) on noisy Pascal-VOC 2012

| Metrics | Methods | Sym. 10% | Sym. 30% | Sym. 50% | Asym. 10% | Asym. 30% | Asym. 50% |
|---|---|---|---|---|---|---|---|
| mAP ↑ | Standard | 71.33±1.02 | 66.08±0.68 | 50.30±1.10 | 74.05±1.57 | 61.60±0.38 | 49.61±0.23 |
|  | ASL | 74.18±1.16 | 68.43±0.23 | 49.72±1.08 | 74.90±1.53 | 62.66±0.61 | 48.82±1.07 |
|  | LRAD-R | 75.94±0.11 | 71.69±0.54 | 70.39±0.47 | 75.97±1.51 | 71.41±1.20 | 63.68±1.13 |
|  | HLC | 77.85±1.48 | 72.84±1.82 | 69.03±1.34 | 75.17±0.94 | 70.62±0.77 | 64.27±1.62 |
|  | LSNPC-R | 80.52±1.96 | 75.07±0.34 | 73.26±0.43 | 77.89±0.13 | 72.10±1.69 | 65.08±1.26 |
|  | CAD-R$^\dagger$ | 81.89±0.21 | 77.02±0.94 | 75.53±0.17 | 80.49±0.27 | 75.37±0.39 | 68.36±0.89 |
|  | LRAD-V | 83.86±0.34 | 81.15±0.39 | 80.38±1.88 | 81.67±0.78 | 79.81±1.67 | 72.91±1.14 |
|  | LSNPC-V | 85.09±1.41 | 82.16±0.60 | 83.17±1.35 | 81.42±0.60 | 80.88±1.82 | 73.75±1.49 |
|  | CAD-V$^\dagger$ | 89.27±0.97 | 88.99±0.69 | 88.31±0.72 | 89.17±1.03 | 82.12±0.25 | 76.63±0.59 |
| OF1 ↑ | Standard | 66.09±0.28 | 64.28±0.79 | 49.95±1.06 | 68.32±1.45 | 58.21±1.74 | 43.67±1.43 |
|  | ASL | 68.74±0.71 | 65.89±0.77 | 49.50±1.18 | 69.12±1.12 | 56.18±1.36 | 45.37±0.24 |
|  | LRAD-R | 70.36±0.66 | 61.60±2.00 | 55.76±1.10 | 70.11±0.35 | 57.49±0.29 | 46.91±0.12 |
|  | HLC | 72.14±1.73 | 70.19±0.11 | 67.83±1.05 | 72.37±0.77 | 65.56±0.65 | 60.64±0.89 |
|  | LSNPC-R | 74.61±1.08 | 68.56±0.24 | 68.01±0.73 | 72.92±1.39 | 64.04±0.66 | 58.01±0.11 |
|  | CAD-R$^\dagger$ | 75.88±0.57 | 72.27±0.77 | 69.80±0.94 | 74.31±0.47 | 65.71±0.71 | 61.60±0.85 |
|  | LRAD-V | 74.72±0.39 | 71.33±1.14 | 70.07±0.64 | 74.37±1.17 | 61.53±1.22 | 59.51±1.29 |
|  | LSNPC-V | 77.83±0.13 | 72.61±0.87 | 71.76±1.10 | 75.13±0.84 | 65.05±1.14 | 61.44±1.92 |
|  | CAD-V$^\dagger$ | 82.72±0.80 | 77.33±0.61 | 75.22±0.89 | 82.59±0.83 | 70.31±1.17 | 66.78±1.28 |
| CF1 ↑ | Standard | 66.74±1.31 | 57.74±1.05 | 44.22±1.18 | 68.98±0.83 | 56.40±1.22 | 43.36±1.79 |
|  | ASL | 69.42±0.80 | 59.81±1.38 | 48.90±0.25 | 69.78±0.63 | 57.68±0.48 | 46.08±1.73 |
|  | LRAD-R | 71.07±0.59 | 60.65±0.24 | 54.70±1.47 | 70.78±0.12 | 59.36±1.05 | 47.64±1.76 |
|  | HLC | 72.85±0.65 | 68.54±1.73 | 64.37±0.68 | 71.89±0.83 | 65.42±0.74 | 59.86±1.75 |
|  | LSNPC-R | 75.35±1.77 | 65.61±1.10 | 64.12±0.66 | 73.61±1.48 | 60.81±0.60 | 58.14±0.99 |
|  | CAD-R$^\dagger$ | 76.63±0.57 | 68.62±0.11 | 64.86±0.73 | 74.98±0.95 | 65.47±0.58 | 60.56±1.41 |
|  | LRAD-V | 74.48±1.49 | 70.54±1.28 | 64.07±0.40 | 73.09±0.71 | 62.93±0.37 | 60.77±0.79 |
|  | LSNPC-V | 79.62±0.89 | 71.80±0.58 | 70.69±0.91 | 75.85±0.39 | 65.99±0.46 | 63.66±1.38 |
|  | CAD-V$^\dagger$ | 83.54±0.13 | 78.64±0.77 | 71.23±0.21 | 83.38±0.29 | 72.44±0.94 | 67.49±1.07 |

## 5.2 Comparison with SOTA methods

The results on noisy Pascal-VOC 2007, Pascal-VOC 2012, MS-COCO and real-world noisy dataset NUS-WIDE are shown in Table 1, Table 2, Table 3 and Table 4, respectively. Our method (**denoted by †**) consistently outperforms SOTA methods across synthetic and real-world noisy dataset. We analyze our experimental results in two groups based on different pre-trained encoders.

**Compared with the Common group.** First, our method consistently outperforms the Standard across all noise settings. For instance, on Pascal-VOC 2007 with 30% symmetric noise, our method improves mAP by 11% and OF1 by 8%, demonstrating the necessity of handling noise in NML problem. Second, when combined with a ResNet pre-trained encoder, our approach maintains a stable advantage over SOTA methods such as LSNPC-R and HLC, e.g., on Pascal-VOC 2012 with 50% noise, our method surpasses SOTA by an average of 2%. Notably, compared to mAP, the OF1 and the CF1 scores of CAD-R show less stability. In the high-noise setting of the Pascal-VOC 2007 dataset, the CF1 score is slightly lower than HLC, but still competitive. We attribute this to the bias in neighborhood proxy estimation and the noise leakage from the meta-subset, and analyzes the limitations of the CAD-R version in high-noise environments (see Appendix F). Therefore, we recommend using the CAD-V version, which offers better performance and stability.

**Compared with the ViT group.** CAD-V exhibits even greater improvements than *Common group*. On MS-COCO, it outperforms LRAD-V and LSNPC-V by an average of 8% and 4%, respectively, highlighting its superior ability to leverage the same prior information more effectively. Compared to the Standard method, our method achieves over a 30% improvement in high-noise settings (50% noise rate), validating its robustness. Additionally, we compare the CAD-V version with the respective state-of-the-art methods on the real-world noisy dataset NUS-WIDE, and the results are provided in the Table 4, further demonstrating that our method can well handle practical scenes. As shown in Figure 2, our method not only corrects erroneous labels (e.g., the second image in the first row, where 'tiger' is corrected to 'cat') but also generates missing labels (e.g., the third image in the first row, where 'plants' and 'rocks' are added). This underscores our method's practical effectiveness in real-world settings and its potential to extend from noisy label learning to partial label learning.

Table 3: The mean and standard deviation of results (%) on noisy MS-COCO

| Metrics | Methods | Sym. 10% | Sym. 30% | Sym. 50% | Asym. 10% | Asym. 30% | Asym. 50% |
|---|---|---|---|---|---|---|---|
| mAP ↑ | Standard | 59.27±0.86 | 54.90±0.73 | 40.75±0.75 | 61.61±0.69 | 55.48±0.62 | 36.85±1.96 |
| | ASL | 61.64±1.10 | 56.85±0.61 | 40.29±0.74 | 62.32±1.52 | 56.44±0.32 | 39.22±0.48 |
| | LRAD-R | 63.11±0.45 | 59.55±1.42 | 57.03±0.78 | 63.21±0.94 | 64.31±1.19 | 45.44±1.74 |
| | HLC | 64.69±1.10 | 61.34±1.95 | 57.53±1.10 | 64.20±0.57 | 63.60±0.28 | 49.25±0.55 |
| | LSNPC-R | 66.91±1.86 | 62.36±0.80 | 59.36±0.86 | 65.73±1.45 | 64.94±1.56 | 59.07±1.21 |
| | CAD-R[†] | 68.05±0.87 | 63.99±0.80 | 61.20±0.32 | 66.97±0.59 | 67.88±0.89 | 60.05±0.82 |
| | LRAD-V | 69.69±1.07 | 69.91±1.90 | 66.75±0.36 | 67.95±1.66 | 69.78±0.82 | 68.91±0.98 |
| | LSNPC-V | 70.71±1.35 | 68.25±1.35 | 67.39±0.54 | 67.75±1.19 | 68.84±1.15 | 67.76±0.64 |
| | CAD-V[†] | 74.18±0.11 | 73.93±0.62 | 71.55±0.75 | 74.19±0.65 | 73.96±0.57 | 69.56±0.32 |
| OF1 ↑ | Standard | 57.43±0.65 | 58.51±0.64 | 42.59±0.33 | 59.46±0.66 | 54.57±0.27 | 38.84±1.84 |
| | ASL | 59.74±1.48 | 59.97±1.44 | 42.21±0.64 | 60.15±1.79 | 59.13±0.36 | 41.46±0.98 |
| | LRAD-R | 61.15±1.82 | 56.08±1.19 | 46.07±1.18 | 61.02±0.20 | 60.42±0.46 | 44.94±0.30 |
| | HLC | 62.69±0.80 | 63.29±1.78 | 59.01±0.72 | 61.97±0.11 | 60.60±1.09 | 58.09±0.35 |
| | LSNPC-R | 64.84±1.16 | 62.40±1.74 | 57.99±1.31 | 63.46±1.55 | 59.04±1.50 | 55.61±0.86 |
| | CAD-R[†] | 65.95±0.24 | 63.78±0.82 | 59.52±0.74 | 64.67±0.39 | 61.56±0.61 | 59.01±0.65 |
| | LRAD-V | 66.41±1.80 | 65.75±1.36 | 59.46±0.67 | 64.59±1.95 | 63.99±0.41 | 55.96±1.06 |
| | LSNPC-V | 67.64±1.68 | 64.27±0.34 | 61.19±1.07 | 65.39±0.37 | 64.05±0.34 | 58.85±1.17 |
| | CAD-V[†] | 71.89±0.91 | 70.39±0.45 | 64.14±0.68 | 71.88±0.61 | 69.31±0.90 | 63.97±0.82 |
| CF1 ↑ | Standard | 56.42±0.65 | 50.24±0.94 | 38.65±0.61 | 58.45±0.16 | 53.19±1.10 | 37.36±0.92 |
| | ASL | 58.68±1.55 | 52.03±1.97 | 42.75±1.18 | 59.13±1.52 | 54.40±1.18 | 40.89±1.25 |
| | LRAD-R | 60.07±1.80 | 54.51±1.53 | 47.81±1.14 | 59.97±1.44 | 57.87±0.57 | 42.27±0.70 |
| | HLC | 61.59±0.24 | 56.15±0.13 | 56.26±0.25 | 60.91±1.08 | 59.81±0.97 | 54.81±1.13 |
| | LSNPC-R | 63.69±0.68 | 57.08±1.47 | 56.05±0.13 | 62.37±0.53 | 57.35±0.90 | 53.59±1.51 |
| | CAD-R[†] | 64.78±0.54 | 58.57±0.50 | 56.69±0.90 | 63.53±0.91 | 59.86±0.95 | 54.85±0.75 |
| | LRAD-V | 63.34±0.11 | 59.99±1.06 | 56.75±1.76 | 62.47±0.11 | 61.18±1.16 | 51.58±0.89 |
| | LSNPC-V | 67.30±1.47 | 62.47±1.88 | 58.29±1.39 | 64.27±1.27 | 62.24±0.70 | 56.48±1.83 |
| | CAD-V[†] | 70.62±0.25 | 68.42±0.59 | 62.26±0.93 | 70.65±0.70 | 68.32±0.48 | 59.88±1.06 |

Table 4: Comparison of our method's result (%) to SOTA on the NUS-WIDE dataset.

| Metrics | Standard [33] | ASL [35] | HLC [27] | LRAD-V [31] | LSNPC-V [28] | CAD-V[†] |
|---|---|---|---|---|---|---|
| mAP ↑ | 59.21 | 63.92 | 63.14 | 63.95 | 64.37 | **65.13** |
| OF1 ↑ | 71.52 | 75.03 | 74.68 | 72.13 | 74.92 | **75.58** |
| CF1 ↑ | 57.77 | 62.69 | 62.87 | 60.53 | 63.43 | **63.68** |

## 5.3 Ablation Studies

We conducted ablation experiments on Pascal-VOC 2012 and MS-COCO to assess the effectiveness of three modules: neighborhood label proxy ($\bar{y}$), co-occurrence-aware (CA) strategy, and diffusion model (DM). As shown in Table 5, disabling all three modules results in the Standard baseline. Enabling only the DM approximates the LRAD model without neighborhood retrieval, causing a 9% performance drop. Conversely, disabling the DM reduces the method to a non-parametric neighborhood estimation, performing worse than most other settings. Notably, the whole framework's performance is only closely matched when DM is combined with either $\bar{y}$ or CA, highlighting the importance of the diffusion model. Additional ablation studies on pre-trained $f_p$, neighborhood proxy estimation performance, the impact of neighbor $K$ value, and training efficiency analysis are detailed in Appendices E, F, G, and H, respectively.

Table 6 further demonstrates that, compared to neighborhood estimation results under different pretrained encoders, the diffusion paradigm consistently plays a critical role in CAD. For example, when ResNet50—with relatively weaker representation capacity—is used as the pretrained encoder, its neighborhood estimation performance falls below that of the baseline HLC. Even with such coarse estimations, CAD is capable of refining the generative feature-label mapping and consistently achieving performance improvements beyond the label estimation stage. Moreover, the CAD model with ViT-14/L still outperforms the baseline, whereas the neighborhood label estimation alone (based on ViT-14/L) performs worse than the baseline across key metrics like OF1 and CF1. This clearly highlights the indispensable role of the diffusion architecture in enhancing performance beyond what weak feature spaces can offer.

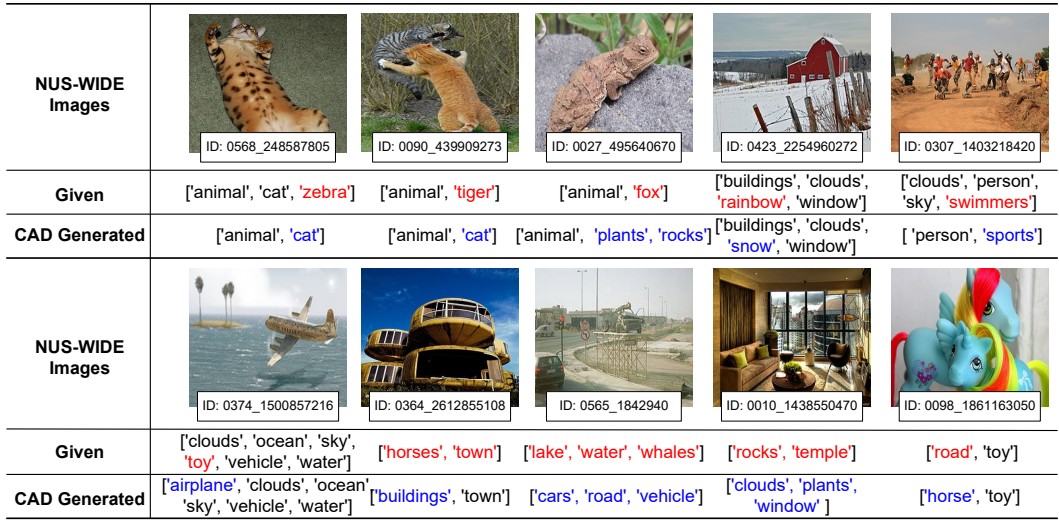

| | | | | | |
|---|---|---|---|---|---|
| **NUS-WIDE Images** | ID: 0568_248587805 | ID: 0090_439909273 | ID: 0027_495640670 | ID: 0423_2254960272 | ID: 0307_1403218420 |
| **Given** | ['animal', 'cat', 'zebra'] | ['animal', 'tiger'] | ['animal', 'fox'] | ['buildings', 'clouds', 'rainbow', 'window'] | ['clouds', 'person', 'sky', 'swimmers'] |
| **CAD Generated** | ['animal', 'cat'] | ['animal', 'cat'] | ['animal', 'plants', 'rocks'] | ['buildings', 'clouds', 'snow', 'window'] | [ 'person', 'sports'] |
| **NUS-WIDE Images** | ID: 0374_1500857216 | ID: 0364_2612855108 | ID: 0565_1842940 | ID: 0010_1438550470 | ID: 0098_1861163050 |
| **Given** | ['clouds', 'ocean', 'sky', 'toy', 'vehicle', 'water'] | ['horses', 'town'] | ['lake', 'water', 'whales'] | ['rocks', 'temple'] | ['road', 'toy'] |
| **CAD Generated** | ['airplane', 'clouds', 'ocean', 'sky', 'vehicle', 'water'] | ['buildings', 'town'] | ['cars', 'road', 'vehicle'] | ['clouds', 'plants', 'window' ] | ['horse', 'toy'] |

Figure 2: True noisy multi-label examples from the NUS-WIDE dataset and the prediction results from the proposed CAD model. The noisy items in the ground truth label set are highlighted in red, while the successfully corrected items in the CAD model's generated label set are highlighted in blue.

Table 5: Ablation study results with 30% noise, reporting the average OF1 scores (%).

| Modules | | | Pascal-VOC 2012 | | | | MS-COCO | | | |
|---|---|---|---|---|---|---|---|---|---|---|
| $\bar{y}$ | CA | DM | Sym. | Gap ↓ | Asym. | Gap ↓ | Sym. | Gap ↓ | Asym. | Gap ↓ |
| ✗ | ✗ | ✗ | 64.28 | 13.05 | 58.21 | 12.10 | 58.51 | 11.88 | 54.57 | 14.74 |
| ✗ | ✗ | ✓ | 70.31 | 7.02 | 60.52 | 9.97 | 64.75 | 5.64 | 62.98 | 6.33 |
| ✓ | ✗ | ✓ | 74.29 | 3.04 | 68.41 | 1.90 | 67.35 | 3.04 | 66.58 | 2.73 |
| ✗ | ✓ | ✓ | 75.43 | 1.90 | 67.99 | 2.32 | 68.13 | 2.26 | 65.97 | 3.34 |
| ✓ | ✓ | ✗ | 71.66 | 5.67 | 64.87 | 5.44 | 63.47 | 6.92 | 62.53 | 6.78 |
| ✓ | ✓ | ✓ | **77.33** | N/A | **70.31** | N/A | **70.39** | N/A | **69.31** | N/A |

Table 6: Comparison of the two phased results of label pre-estimation ($\bar{y}$) and CAD with different pre-trained encoders. HLC is presented as the baseline in the first row, while the best and second-best metrics are highlighted in red and blue, respectively.

| Methods | Pre-trained Encoders | VOC 2007-sym. 50% | | | VOC 2007-Asym. 50% | | | VOC 2012-Sym. 50% | | | VOC 2012-Asym. 50% | | |
|---|---|---|---|---|---|---|---|---|---|---|---|---|---|
| | | mAP | OF1 | CF1 | mAP | OF1 | CF1 | mAP | OF1 | CF1 | mAP | OF1 | CF1 |
| HLC | N/A | 68.03 | 66.62 | 63.65 | 59.09 | 58.59 | 56.51 | 69.03 | 67.83 | 64.37 | 64.27 | 60.64 | 59.86 |
| $\bar{y}$ | ResNet50 | 63.65 | 47.42 | 44.53 | 45.36 | 31.62 | 37.88 | 65.88 | 39.69 | 35.31 | 59.56 | 32.62 | 31.61 |
| CAD | ResNet50 | 72.37 | 67.19 | 64.13 | 61.88 | 59.51 | 56.17 | 75.53 | 69.80 | 64.86 | 68.36 | 61.60 | 60.56 |
| $\bar{y}$ | ViT-14/L | 82.20 | 53.21 | 55.43 | 52.38 | 39.99 | 46.57 | 82.27 | 48.64 | 47.79 | 52.17 | 44.23 | 48.68 |
| CAD | ViT-14/L | 84.61 | 72.41 | 69.33 | 69.37 | 64.51 | 62.74 | 88.31 | 75.22 | 71.24 | 76.63 | 66.78 | 67.49 |

## 6 Conclusion

In this work, we advocate for deep generative perspective to achieve robust multi-label classification, offering a novel insight. We use powerful diffusion models to reformulate NML within a probabilistic denoising label learning and robust inference paradigm, proposing the CAD model. We enhance the reliability of the target distribution in the forward process through neighborhood proxy estimation in the latent feature space, while constraining erroneous generation directions in the reverse process using label co-occurrence rates. To the best of our knowledge, this is the first application of diffusion models to the NML problem. The proposed method achieves SOTA performance on both synthetic and real-world noisy datasets, highlighting its strong potential in this domain.

## Acknowledgments

This work was supported in part by the National Natural Science Foundation of China under Grants U21A20513, 62476157, 62576201, and 62276161.

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

# Appendix

## A  Unconditional Multi-label Diffusion Models

In this section, we model unconditional label generation following the theoretical framework of denoising diffusion probabilistic models (DDPM) [32]. We start by defining the data distribution $y_0 \sim q(y_0)$ and a Markovian forward (diffusion) process $q$, which progressively adds Gaussian noise $\epsilon \sim \mathcal{N}(0, \mathbf{I})$ to the labels based on diffusion strength $\beta_t$, diffusing label $y_0$ to $y_T$:

$$q(y_t \mid y_{t-1}) = \mathcal{N}\left(y_t; \sqrt{1 - \beta_t} y_{t-1}, \beta_t \mathbf{I}\right), \tag{10}$$

According to the derivation of the Markov chain, the joint probability distribution from $y_0$ to $y_t$ can be computed, which is also a Gaussian distribution:

$$q(y_t \mid y_0) = \mathcal{N}\left(y_t; \sqrt{\bar{\alpha}_t} y_0, (1 - \bar{\alpha}_t) \mathbf{I}\right) \tag{11}$$

$$= \sqrt{\bar{\alpha}_t} y_0 + \sqrt{1 - \bar{\alpha}_t} \epsilon, \quad \epsilon \sim \mathcal{N}(0, \mathbf{I}). \tag{12}$$

where $\alpha_t := 1 - \beta_t$ and $\bar{\alpha}_t := \prod_{s=0}^{t} \alpha_s$. Under Bayes' theorem, one finds that the posterior distribution of the forward process can be represented as a Gaussian distribution with mean $\tilde{\mu}_t(y_t, y_0)$ and variance $\tilde{\beta}_t$:

$$q(y_{t-1} \mid y_t, y_0) = \mathcal{N}\left(y_{t-1}; \tilde{\mu}(y_t, y_0), \tilde{\beta}_t \mathbf{I}\right), \tag{13}$$

$$\tilde{\mu}_t(y_t, y_0) = \frac{\sqrt{\bar{\alpha}_{t-1}} \beta_t}{1 - \bar{\alpha}_t} y_0 + \frac{\sqrt{\alpha_t}(1 - \bar{\alpha}_{t-1})}{1 - \bar{\alpha}_t} y_t, \tag{14}$$

$$\tilde{\beta}_t = \frac{1 - \bar{\alpha}_{t-1}}{1 - \bar{\alpha}_t} \beta_t, \tag{15}$$

In the reverse process, we naturally employ a neural network to perform Markovian inference from $q(y_T)$, fitting the posterior distribution $q(y_{t-1} \mid y_t)$ through to $y_0$. Since the variance of the posterior Gaussian distribution depends only on the diffusion strength corresponding to time $t$, the neural network focuses primarily on predicting and learning the mean:

$$p_\theta(y_{t-1} \mid y_t) := \mathcal{N}\left(y_{t-1}; \mu_\theta(y_t, t), \tilde{\beta}_t \mathbf{I}\right). \tag{16}$$

To train this model such that $p_\theta(y_0)$ closely approximates $q(y_0)$, we optimize the variational lower bound (VLB):

$$L_{VLB} := L_0 + L_1 + \ldots + L_{T-1} + L_T, \tag{17}$$

$$L_0 := -\log p_\theta(y_0 \mid y_1), \tag{18}$$

$$L_{t-1} := D_{KL}(q(y_{t-1} \mid y_t, y_0) \,\|\, p_\theta(y_{t-1} \mid y_t)), \tag{19}$$

$$L_T := D_{KL}(q(y_T \mid y_0) \,\|\, p(y_T)). \tag{20}$$

However, Ho et al. [32] found that directly predicting the mean was not effective, so they reparameterized the estimation target, constraining the neural network to estimate the actual noise of the diffusion process and simplified the loss function:

$$L_{\text{simple}} := E_{t \sim [1,T], y_0 \sim q(y_0), \epsilon \sim \mathcal{N}(0,\mathbf{I})} \left[\| \epsilon - \epsilon_\theta(y_t, t) \|^2\right]. \tag{21}$$

During inference, we utilize the label noise estimated by the model, restoring the mean of the label at time $t - 1$ to retrieve its distribution form, and then gradually performing random sampling until generating the label $y_0$:

$$y_{t-1} = \frac{1}{\sqrt{\alpha_t}} \left(y_t - \frac{1 - \alpha_t}{\sqrt{1 - \bar{\alpha}_t}} \epsilon_\theta(y_t, t)\right) + \sqrt{\tilde{\beta}_t} \mathbf{z}, \tag{22}$$

where $\mathbf{z} \sim \mathcal{N}(0, \mathbf{I})$. Unfortunately, such a label generation process does not incorporate any conditional information, thus for instances with same features, it can generate a variety of labels. This characteristic makes the diffusion model impractical for classification tasks where distinct label assignment is required.

# B  Conditional Multi-label Diffusion Models and Feature Embedding

In this section, our main goal is to derive the posterior distribution of the forward diffusion process when features are used as controlling conditions and to analyze its differences and connections with the unconditional posterior distribution. Following the method used in DDPM [32], We first define a conditional Markov process $\hat{q}$ (where $q$ represents the corresponding unconditional process) in which Gaussian noise $\epsilon \sim \mathcal{N}(0, \mathbf{I})$ is incrementally added to the labels for diffusion. The addition of noise remains the same whether features are conditioned or not, leading to the following definition:

$$\hat{q}(y_0) := q(y_0), \tag{23}$$

$$\hat{q}(y_t \mid y_{t-1}, x) := q(y_t \mid y_{t-1}), \tag{24}$$

$$\hat{q}(y_{1:T} \mid y_0, x) := \prod_{t=1}^{T} \hat{q}(y_t \mid y_{t-1}, x). \tag{25}$$

With knowledge of the forward process's prior distribution, we can derive the prior distribution of $\hat{q}$:

$$\hat{q}(y_t \mid y_{t-1}) = \int_x \hat{q}(y_t, x \mid y_{t-1}) \, dx \tag{26}$$

$$= \int_x \hat{q}(y_t \mid y_{t-1}, x) \, \hat{q}(x \mid y_{t-1}) \, dx \tag{27}$$

$$= \int_x q(y_t \mid y_{t-1}) \, \hat{q}(x \mid y_{t-1}) \, dx \tag{28}$$

$$= q(y_t \mid y_{t-1}) \int_x \hat{q}(x \mid y_{t-1}) \, dx \tag{29}$$

$$= q(y_t \mid y_{t-1}) \tag{30}$$

$$= \hat{q}(y_t \mid y_{t-1}, x), \tag{31}$$

which indicates that conditions do not affect the prior distribution in the forward process. Similarly, we can derive the joint distribution of $\hat{q}$:

$$\hat{q}(y_{1:T} \mid y_0) = \int_x \hat{q}(y_{1:T}, x \mid y_0) \, dx \tag{32}$$

$$= \int_x \hat{q}(x \mid y_0) \, \hat{q}(y_{1:T} \mid y_0, x) \, dx \tag{33}$$

$$= \int_x \hat{q}(x \mid y_0) \prod_{t=1}^{T} \hat{q}(y_t \mid y_{t-1}, x) dx \tag{34}$$

$$= \int_x \hat{q}(x \mid y_0) \prod_{t=1}^{T} q(y_t \mid y_{t-1}) dx \tag{35}$$

$$= \prod_{t=1}^{T} q(y_t \mid y_{t-1}) \int_x \hat{q}(x \mid y_0) \, dx \tag{36}$$

$$= \prod_{t=1}^{T} q(y_t \mid y_{t-1}) \tag{37}$$

$$= q(y_{1:T} \mid y_0). \tag{38}$$

Based on this result, we can further derive the marginal distribution of $\hat{q}$:

$$\hat{q}\left(y_{t}\right)=\int_{y_{0:t-1}}\hat{q}\left(y_{0},\ldots,y_{t}\right)dy_{0:t-1} \tag{39}$$

$$=\int_{y_{0:t-1}}\hat{q}\left(y_{0}\right)\hat{q}\left(y_{1},\ldots,y_{t}\mid y_{0}\right)dy_{0:t-1} \tag{40}$$

$$=\int_{y_{0:t-1}}q\left(y_{0}\right)q\left(y_{1},\ldots,y_{t}\mid y_{0}\right)dy_{0:t-1} \tag{41}$$

$$=\int_{y_{0:t-1}}q\left(y_{0},\ldots,y_{t}\right)dy_{0:t-1} \tag{42}$$

$$=q\left(y_{t}\right). \tag{43}$$

Using the prior and marginal distributions, we can demonstrate that the unconditional posterior distribution aligns with $q$:

$$\hat{q}\left(y_{t-1}\mid y_{t}\right)=\frac{\hat{q}\left(y_{t-1},y_{t}\right)}{\hat{q}\left(y_{t}\right)} \tag{44}$$

$$=\frac{\hat{q}\left(y_{t}\mid y_{t-1}\right)\hat{q}\left(y_{t-1}\right)}{\hat{q}\left(y_{t}\right)} \tag{45}$$

$$=\frac{q\left(y_{t}\mid y_{t-1}\right)q\left(y_{t-1}\right)}{q\left(y_{t}\right)} \tag{46}$$

$$=\frac{q\left(y_{t-1},y_{t}\right)}{q\left(y_{t}\right)} \tag{47}$$

$$=q\left(y_{t-1}\mid y_{t}\right). \tag{48}$$

By incorporating features as posterior conditions, we estimate the posterior distribution of the conditional forward process using Bayes' rule:

$$\hat{q}\left(y_{t-1}\mid x\right)=\frac{\hat{q}\left(y_{t-1}\right)\hat{q}\left(x\mid y_{t-1}\right)}{\hat{q}(x)}. \tag{49}$$

Continuing, by adding the known distribution $y_t$ as a condition for generation, we can obtain:

$$\hat{q}\left(y_{t-1}\mid y_{t},x\right)=\frac{\hat{q}\left(y_{t-1}\mid y_{t}\right)\hat{q}\left(x\mid y_{t-1},y_{t}\right)}{\hat{q}\left(x|y_{t}\right)} \tag{50}$$

$$=\frac{q\left(y_{t-1}\mid y_{t}\right)\hat{q}\left(x\mid y_{t-1},y_{t}\right)}{\hat{q}\left(x|y_{t}\right)} \tag{51}$$

$$=\frac{q\left(y_{t-1}\mid y_{t}\right)\hat{q}\left(x\mid y_{t-1}\right)}{\hat{q}\left(x|y_{t}\right)} \tag{52}$$

$$=q\left(y_{t-1}\mid y_{t}\right)e^{\log\hat{q}(x|y_{t-1})-\log\hat{q}(x|y_{t})}, \tag{53}$$

where the derivation of $\hat{q}\left(x\mid y_{t-1},y_{t}\right)=\hat{q}\left(x\mid y_{t-1}\right)$ from Eq. (51) to Eq. (52) is as follows:

$$\hat{q}\left(x\mid y_{t-1},y_{t}\right)=\hat{q}\left(y_{t}\mid y_{t-1},x\right)\frac{\hat{q}\left(x\mid y_{t-1}\right)}{\hat{q}\left(y_{t}\mid y_{t-1}\right)} \tag{54}$$

$$=\hat{q}\left(y_{t}\mid y_{t-1}\right)\frac{\hat{q}\left(x\mid y_{t-1}\right)}{\hat{q}\left(y_{t}\mid y_{t-1}\right)} \tag{55}$$

$$=\hat{q}\left(x\mid y_{t-1}\right). \tag{56}$$

We note that the term $e^{-\log\hat{q}(x|y_{t})}$ in Eq. (53) is independent of the distribution of $y_{t-1}$, thus we set this part as a constant $A$:

$$\hat{q}\left(y_{t-1}\mid y_{t},x\right)=A\cdot q\left(y_{t-1}\mid y_{t}\right)e^{\log\hat{q}(x|y_{t-1})}, \tag{57}$$

where $q\left(y_{t-1}\mid y_{t}\right)$ is the unconditional posterior distribution of the diffusion process, modeled as a Gaussian distribution with mean $\tilde{\mu}_{t}\left(y_{t},y_{0}\right)$ and variance $\tilde{\beta}_{t}$ in Eq. (14) and Eq. (15), respectively. Simplifying the covariance from the probability density formula, we can get:

$$\hat{q}\left(y_{t-1}\mid y_{t},x\right)\propto e^{-\|y_{t-1}-\tilde{\mu}_{t}\|^{2}/2\tilde{\beta}_{t}+\log\hat{q}(x|y_{t-1})}. \tag{58}$$

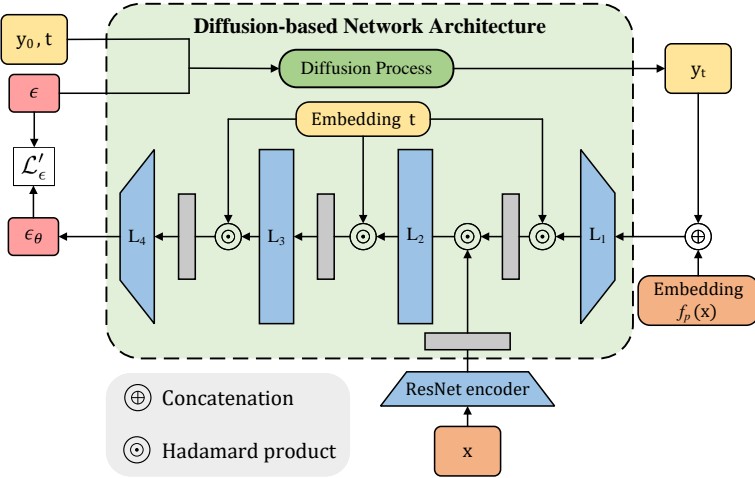

Figure B.1: The network architecture of the conditional multi-label diffusion model. Blue components represent trainable network layers, while gray components represent normalization activation layers.

Given that the number of time steps $T$ in the diffusion process is large enough and the diffusion coefficient $\beta_t$ is small enough, the variance of the distribution $\hat{q}(y_{t-1} \mid y_t)$ is sufficiently small and concentrated near $\tilde{\mu}_t$. We perform a Taylor expansion around $y_{t-1} = \tilde{\mu}_t$ for $\log \hat{q}(x \mid y_{t-1})$ up to the first derivative, for simplicity, we let $\nabla_{y_{t-1}} \log \hat{q}(x \mid y_{t-1})|_{y_{t-1}=\tilde{\mu}_t} = g$, which is essentially the gradient of the distribution at that point.:

$$\log \hat{q}(x \mid y_{t-1}) \propto \log \hat{q}(x \mid y_{t-1})|_{y_{t-1}=\tilde{\mu}_t} + (y_{t-1} - \tilde{\mu}_t)g + \mathbf{o}(y_{t-1}). \tag{59}$$

Thus, the posterior distribution can be estimated as:

$$\hat{q}(y_{t-1} \mid y_t, x) \propto e^{-\|y_{t-1}-\tilde{\mu}_t\|^2/2\tilde{\beta}_t+(y_{t-1}-\tilde{\mu}_t)g+C_1} \tag{60}$$

$$\propto e^{-(\|y_{t-1}-\tilde{\mu}_t-\tilde{\beta}_t g\|^2)/2\tilde{\beta}_t+C_2} \tag{61}$$

$$= \mathcal{N}\left(y_{t-1}; \hat{\mu}_t, \sigma_t^2 \mathbf{I}\right), \tag{62}$$

where $\hat{\mu}_t = \frac{\beta_t \sqrt{\bar{\alpha}_{t-1}}}{1-\bar{\alpha}_t} y_0 + \frac{(1-\bar{\alpha}_{t-1})\sqrt{\alpha_t}}{1-\bar{\alpha}_t} y_t + \sigma_t^2 \nabla_{y_{t-1}} \log \hat{q}(x \mid y_{t-1})$ and $\sigma_t = \sqrt{\tilde{\beta}_t} = \sqrt{\frac{1-\bar{\alpha}_{t-1}}{1-\bar{\alpha}_t}\beta_t}$. To ensure the correct introduction of conditions, we need to incorporate the decoding gradient into the mean during model prediction. The diffusion model's network architecture, depicted in Figure B.1, consists of a ResNet encoder and a series of feedforward layers. The $L_1$ decoding layer plays a crucial role by contributing the gradient $\nabla_{y_t} \log p(x|y_{t-1})$ as guidance. The network inputs are $(x, y_0)$, randomly sampled $t$ and $\epsilon$, where $y_0$ is transformed into $y_t$ by forward noise addition and then concatenated with $f_p(x)$. After decoding, it merges with the normalized encoding features of ResNet through a Hadamard product, incorporates time positional encoding, and uses a series of feedforward networks, batch normalization, and Softplus activation to predict the noise term $\epsilon_\theta$.

## C   Deterministic Implicit Inference

This section primarily discusses the inference process of the CAD algorithm. Since the diffusion process involves labels and is directed towards classification tasks, it is imperative to reduce the uncertainty in the inference process and expedite it as much as possible, aligning with the method of denoising diffusion implicit models (DDIM) [46]. With a trained conditional diffusion model, we proceed as follows:

$$q(y_t \mid y_0) = \mathcal{N}\left(y_t; \sqrt{\bar{\alpha}_t} y_0, (1-\bar{\alpha}_t)\mathbf{I}\right), \tag{63}$$

$$y_t = \sqrt{\bar{\alpha}_t} y_0 + \sqrt{1-\bar{\alpha}_t}\epsilon, \tag{64}$$

**Algorithm C.1** CAD inference

---

**Input:** Testing set $\mathcal{D} = \{\mathbf{X}\}$
**Output**: $y_0$
1: Sample instance $x \sim \mathcal{D}$, Sample multi-label $y_T \sim \mathcal{N}(0, \mathbf{I})$
2: **for** $s = S$ **to** 1 **do**
3:    $\mathbf{z} \sim \mathcal{N}(0, \mathbf{I})$ if $s > 1$, else $\mathbf{z} = \mathbf{0}$
4:    $y_{\tau_{s-1}} = \sqrt{\bar{\alpha}_{\tau_{s-1}}} \left( \frac{y_{\tau_s} - \sqrt{1 - \bar{\alpha}_{\tau_s}} \cdot \epsilon_\theta^{(\tau_s)}(y_{\tau_s})}{\sqrt{\bar{\alpha}_{\tau_s}}} \right) + \sqrt{1 - \bar{\alpha}_{\tau_{s-1}} - \sigma_s^2} \cdot \epsilon_\theta^{(\tau_s)}(y_{\tau_s}) + \sigma_s \mathbf{z}$
5: **end for**

---

where $\epsilon \sim \mathcal{N}(0, \mathbf{I})$. Similar to DDIM, we define a non-Markovian nature for the forward posterior distribution:

$$q_{\sigma_s}\left(y_{\tau_{s-1}} \mid y_{\tau_s}, y_0\right) = \mathcal{N}\left(y_{\tau_{s-1}}; My_0 + Ny_{\tau_s}, \sigma_s^2 \mathbf{I}\right), \tag{65}$$

where $M$ and $N$ are coefficients to be determined, and $\sigma_s \geq 0$. $\tau$ is a subsequence of $[1, \cdots, T]$, with $\tau_s = T$, e.g., if $T = 1000$ and $S = 10$, then $\tau = [1, 100, \cdots, 900, 1000]$. From the empirical form of the posterior distribution, we have:

$$y_{\tau_{s-1}} = My_0 + Ny_{\tau_s} + \sigma_s \epsilon \tag{66}$$

$$= My_0 + N\left(\sqrt{\bar{\alpha}_s} y_0 + \sqrt{1 - \bar{\alpha}_{\tau_s}} \dot{\epsilon}\right) + \sigma_s \epsilon \tag{67}$$

$$= \left(M + N\sqrt{\bar{\alpha}_{\tau_s}}\right) y_0 + N\sqrt{1 - \bar{\alpha}_{\tau_s}} \dot{\epsilon} + \sigma_s \epsilon, \tag{68}$$

where $\dot{\epsilon}$ and $\epsilon$ are independent and identically distributed Gaussian noises with additivity, hence:

$$y_{\tau_{s-1}} = \left(M + N\sqrt{\bar{\alpha}_{\tau_s}}\right) y_0 + \sqrt{N^2 \left(1 - \bar{\alpha}_{\tau_s}\right) + \sigma_s} \epsilon \tag{69}$$

$$= \sqrt{\bar{\alpha}_{\tau_{s-1}}} y_0 + \sqrt{1 - \bar{\alpha}_{\tau_{s-1}}} \epsilon. \tag{70}$$

By the method of undetermined coefficients, $M$ and $N$ are determined:

$$M = \sqrt{\bar{\alpha}_{\tau_{s-1}}} - \frac{\sqrt{1 - \bar{\alpha}_{\tau_{s-1}} - \sigma_s^2}}{\sqrt{1 - \bar{\alpha}_{\tau_s}}} \cdot \sqrt{\bar{\alpha}_{\tau_s}}, \tag{71}$$

$$N = \frac{\sqrt{1 - \bar{\alpha}_{\tau_{s-1}} - \sigma_s^2}}{\sqrt{1 - \bar{\alpha}_{\tau_s}}}. \tag{72}$$

We reorganize the inference distribution, and since the $y_0$ term is unknown during actual inference, we estimate it using $y_0 = \frac{y_{\tau_s} - \sqrt{1 - \bar{\alpha}_{\tau_s}} \cdot \epsilon_\theta^{(\tau_s)}(y_{\tau_s})}{\sqrt{\bar{\alpha}_{\tau_s}}}$. The detailed inference process is outlined in Algorithm C.1. For classification tasks, following the DDIM approach, we can achieve an implicit probabilistic diffusion model, turning the inference into a deterministic process given $y_T$ by setting $\sigma_s = 0$ [46]. This modification reduces the variability during inference, ensuring more consistent and reliable label predictions crucial for classification accuracy. Additionally, since the inferred labels are in a matrix form, we apply a necessary post-processing step to project the matrix-form labels into a normalized probability distribution, similar to the SoftMax activation used in classification. To obtain the one-hot label vector, we typically set the category with a probability greater than 0.5 to 1.

# D   Experimental Setup and Details

## D.1   Dataset Details

**Synthetic noisy datasets.** Pascal-VOC 2007 and Pascal-VOC 2012 share the same 20 object categories, with an average of 1.5 labels per image. The Pascal-VOC 2007 dataset consists of 5,011 training images and 4,952 test images, while Pascal-VOC 2012 includes 11,540 training images and 10,991 test images. Since the test set labels for Pascal-VOC 2012 are not publicly available, we follow previous studies and use the Pascal- VOC 2007 test set for its evaluation. The MS-COCO dataset contains 82,081 training images and 40,137 test images, covering 80 object categories with an average of 2.9 labels per image. As shown in the Figure D.1, we visualized the noise transition matrix used in the experiments on Pascal-VOC 2007 and Pascal-VOC 2012.

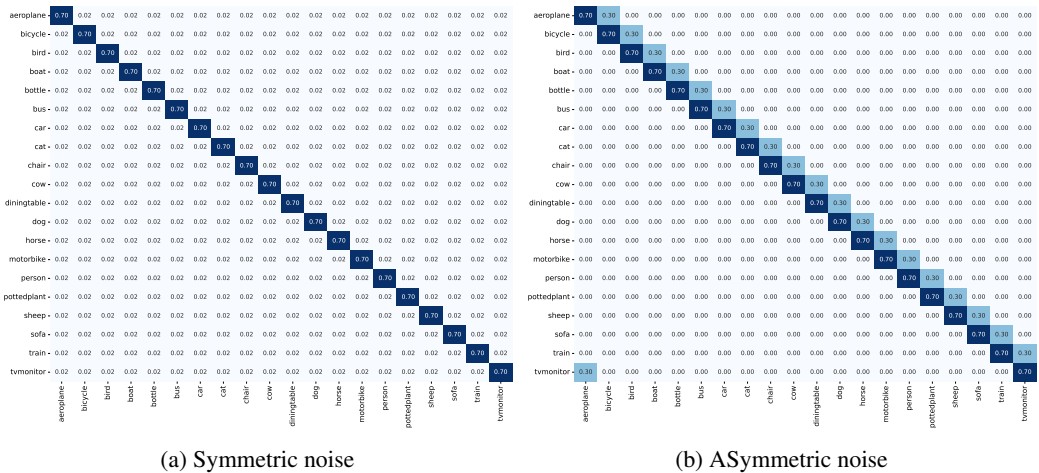

(a) Symmetric noise          (b) ASymmetric noise

Figure D.1: Synthetic noise transition matrices used in our experiments on Pascal-VOC 2007. The noise rate is set to 30%.

**Real-world noisy datasets.** NUS-WIDE dataset [50] originally containing 269,648 images from Flickr and manually annotated with 81 visual concepts, is employed to demonstrate the adaptability of our problem setting to real-world scenarios. Since some download URLs have been deleted, we use the version of the dataset provided in [35]. Our experiments follow the training/testing splits provided by the original dataset. Figure 2 illustrates several real multi-label noisy examples from NUS-WIDE. As shown, our method not only corrects erroneous labels (e.g., the second image in the first row, where 'tiger' is corrected to 'cat') but also generates missing labels (e.g., the third image in the first row, where 'plants' and 'rocks' are added). This underscores our method's practical effectiveness in real-world settings and its potential to extend from noisy label learning to partial label learning.

## D.2 Experimental Details

In our experiments, we configured ResNet50 (depicted as blue ResNet blocks in Figure B.1) as trainable encoders for the diffusion model, where all linear layers have a dimensionality of 1024. We use the Adam optimizer for training 30 epochs with a batch size of 128. The initial learning rate is set to 5e-4, and a half-cycle cosine decay is employed. The images in three datasets resize to $224 \times 224$. The experimental results on the synthetic noisy datasets are averaged over ten independent random trials. Additionally, we used a range of $K$ values from 1 to 100 on the validation set. Experimental results in Appendix G showed that the mAP remained relatively stable for $K$ values between 30 and 60. Based on these results, we inferred that our CAD was relatively insensitive to variations within this range of $K$ values and consequently set the default $K$ value to 50. Due to the increased difficulty of learning from the real-world noisy NUS-WIDE dataset and its higher computational cost, we use ResNet-101 as the trainable encoder for the diffusion model and conduct only a single experiment. The rest of the experimental setup remains the same as for the other three datasets.

## E   More Types of Pre-trained Encoders

We initially conducted research on different settings of the pre-trained encoders $f_p$ and performed ablation experiments comparing our method with Standard, LRAD, and LSNPC. Both the LRAD and LSNPC require the integration of a pre-trained model $f_p$, whereas Standard is a baseline method that does not require any pre-trained models. We selected four pre-trained encoders for our experiments:

- ResNet-50 is a baseline multi-label classifier, we use the ResNet-50 model pre-trained on ImageNet;
- ADDGCN uses a semantic attention module to estimate the content-aware class-label representations for each class from extracted feature map where these representations are fed into a graph convolutional network (GCN) for final multi-label classification;

Table E.1: Results (%) on Pascal-VOC 2012 and MS-COCO with 30% symmetric noise, using various methods with different pre-trained encoders. († denotes our method).

| Method | Pre-trained $f_p$ | Pascal-VOC 2012 | | MS-COCO | |
|---|---|---|---|---|---|
| | | mAP | OF1 | mAP | OF1 |
| Standard | N/A | 66.08 | 64.28 | 54.90 | 58.51 |
| LRAD | ResNet50 | 71.69 | 61.60 | 59.55 | 56.08 |
| LSNPC | ResNet50 | 75.07 | 68.56 | 62.36 | 62.40 |
| CAD† | ResNet50 | 77.02 | 72.27 | 63.99 | 63.78 |
| LRAD | ADDGCN | 73.58 | 64.05 | 62.23 | 58.12 |
| LSNPC | ADDGCN | 78.12 | 71.13 | 64.65 | 64.78 |
| CAD† | ADDGCN | 80.29 | 74.89 | 66.54 | 66.13 |
| LRAD | HLC | 75.13 | 65.36 | 63.55 | 59.30 |
| LSNPC | HLC | 79.58 | 72.54 | 65.82 | 66.04 |
| CAD† | HLC | 81.86 | 76.48 | 67.76 | 67.43 |
| LRAD | ViT-L/14 | 81.15 | 71.33 | 69.91 | 65.75 |
| LSNPC | ViT-L/14 | 82.16 | 72.61 | 68.25 | 64.27 |
| CAD† | ViT-L/14 | **88.99** | **77.33** | **73.93** | **70.39** |

- HLC is a noisy multi-label correction approach built on top of ADDGCN. It uses the ratio between the holistic scores of the example with noisy multi-labels and its variant with predicted labels to correct noisy labels during training. A holistic score measures the instance-label and label dependencies in an example.

- ViT-L/14 is a vision transformer (ViT) model with 306 million parameters, pre-trained on the ImageNet containing over 4 million image-text pairs, providing our framework with exceptional feature extraction capabilities.

We evaluate mAP and OF1 on the Pascal-VOC 2012 and MS-COCO datasets. As shown in Table E.1 Built on feature spaces provided by ADDGCN, designed for multi-label classification, and HLC, optimized for noisy multi-label learning, our CAD method significantly outperforms others, demonstrating superior capability in complex label relationship modeling and noise robustness. Under ADDGCN pretraining, CAD achieves mAP scores of 80.29% and 66.54% on Pascal-VOC 2012 and MS-COCO, respectively, surpassing LSNPC and LRAD, indicating its effectiveness in leveraging graph structures for multi-label modeling. With HLC pretraining, CAD further improves to 81.86% and 67.76% mAP, outperforming all competitors and approaching CAD-V, suggesting its compatibility with other NML-based feature extraction methods. Additionally, under ViT-L/14 pretraining, CAD attains the highest performance (88.99% mAP and 73.93% mAP), reinforcing its efficiency in leveraging prior information. Overall, when the three models use the same pre-trained feature extractor, CAD consistently outperforms the other two methods, indicating that its superior performance is not solely due to the effectiveness of feature extraction.

# F   Effects of Neighborhood Estimation Method

We validate the effectiveness of the neighborhood estimation proxy in the proposed CAD framework on the Pascal-VOC 2007 and Pascal-VOC 2012 datasets. For comparison, we use the label retrieval augmentation (LRA) technique from LRAD as a baseline. As shown in Figure F.1, both methods are negatively impacted by noise intrusion as the noise rate increases. However, our method consistently maintains a significant advantage in mAP. In high-noise environments, however, the OF1 and CF1 scores of our approach rapidly degrade to 0%. This is due to the excessive dispersion of label distributions among neighborhood samples, leading to high entropy in neighborhood proxies, making it difficult to distinguish positive samples. Addressing this limitation of our neighborhood estimation method under high noise conditions will be a focus of future work.

Fortunately, by integrating co-occurrence constraints and diffusion models, our method achieves a 40% performance improvement in high-noise settings (e.g., 50% asymmetric noise), raising OF1 from 30% to 70%. This further validates the importance and necessity of these two additional components.

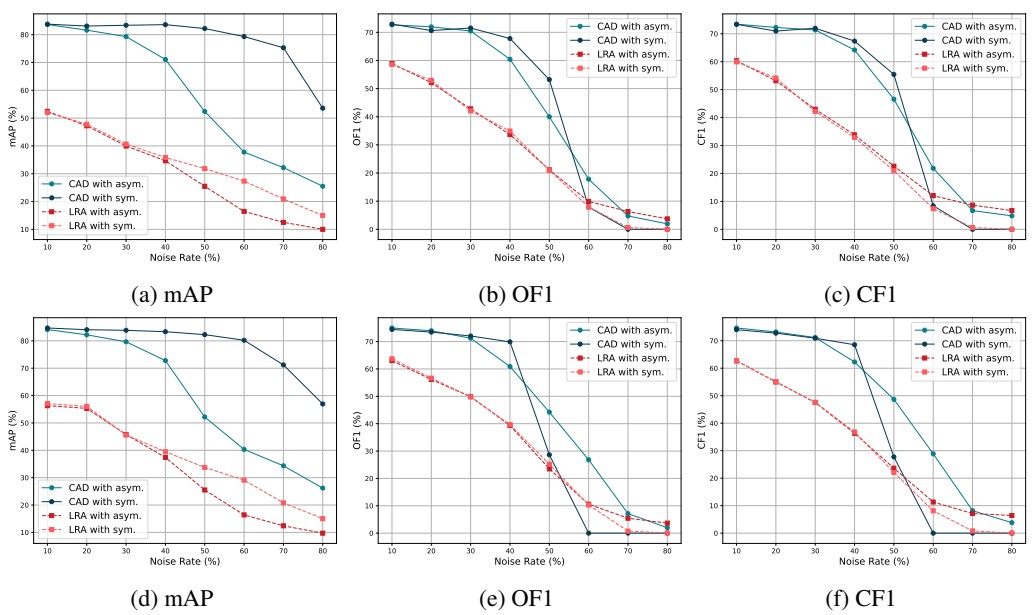

Figure F.1: Comparison of the effectiveness of the neighborhood estimation method in our CAD and the label retrieval augmentation (LRA) in LRAD. Subfigures (a)-(c) present results on the noisy Pascal-VOC 2007 dataset, while Subfigures (d)-(f) show results on the noisy Pascal-VOC 2012 dataset.

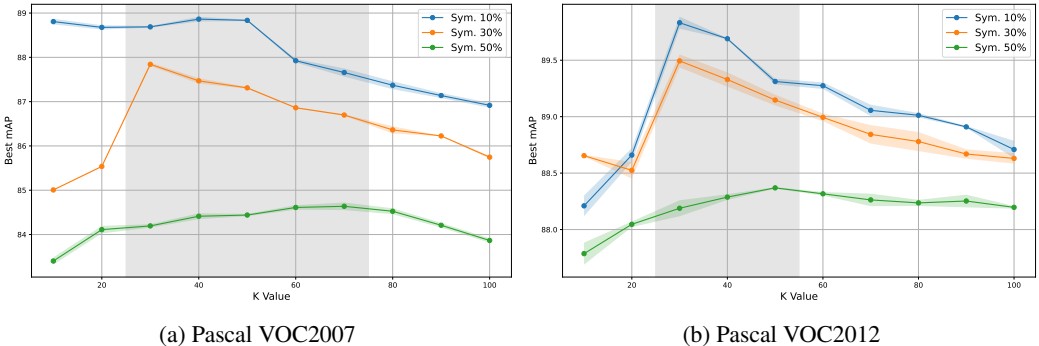

Figure G.1: The effect of different $K$ values on the best mAP (%) of the CAD model under various symmetric noise rates. The gray background highlights the range of $K$ values where the mAP is relatively stable and exhibits robust performance.

## G  Effects of Different *K* Value

To analyze the impact of $K$ values on model performance, we conducted tests on the Pascal-VOC 2007 and Pascal-VOC 2012 datasets under varying Symmetric noise rates with $K$ ranging from 1 to 100. Figure G.1 shows that the CAD's mAP remained relatively stable for $K$ values between 30 and 60. We observe that in low-noise scenarios, a smaller neighborhood radius suffices for accurate estimation, while an excessively large neighborhood can lead to noise leakage. As noise levels increase, the required neighborhood size also grows. This phenomenon can be explained by the need to expand the neighborhood size to stabilize label distributions when noisy information becomes prevalent, thereby improving the model's robustness. However, excessively large values of $K$ increase computational costs without yielding significant improvements in model performance. Considering these factors, we set $K = 50$ as the default setting for CAD, as it achieves both stability and high mAP across various noise conditions.

Table H.1: Test Time (sec) for CAD and Discriminative Models on Different Datasets

| Pascal-VOC (4925 test image) | | | MS-COCO (40,137 test image) | | |
|---|---|---|---|---|---|
| CAD | 8 | Discriminative | 5 | CAD | 173 | Discriminative | 102 |

## H   Training and Inference Efficiency Analysis

The time expenditure comparison in Figure H.1 highlight the significant cost-effectiveness of our CAD. Compared to LSNPC and HLC, CAD achieves optimal performance while saving half of times in training costs, which is groundbreaking. Compared to LRAD, which use similar diffusion model architectures, CAD incurs only about 5 seconds of additional cost. Moreover, its sample preparing process can be pre-computed and cached before training, thereby avoiding the need for repeated calculations. In other words, it is done once, meaning that when dealing with large-scale datasets, the time cost of this process is negligible compared to model training, yet its contribution to enhancing multi-label classification performance is significant.

Table H.1 compares the classification time on the test set between the CAD and discriminative models. Since both Pascal-VOC 2007 and Pascal-VOC 2012 share the test set of Pascal-VOC 2007, we have combined the results. The Pascal-VOC test set consists of 4,925 images, while the MS-COCO test set contains 40,137 images, resulting in a significant increase in testing time. Overall, the diffusion models and discriminative models operate within the same order of magnitude in terms of classification speed. Moreover, classification speed can be further improved with alternative noise scheduling or accelerated sampling strategies, so slow classification does not pose a significant issue.

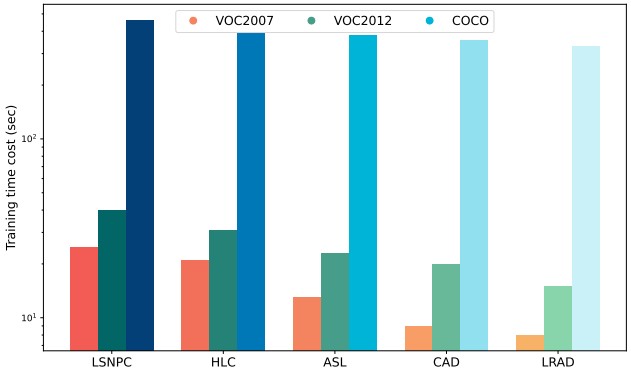

Figure H.1: Time cost (sec) of each method for training one epoch on one NVIDIA A800 GPU.

## I   Limitations and Future Work

In this work, although we demonstrate through experiments that CAD performs strongly on most noisy benchmark datasets, it still has two main limitations: (1) instability under high-noise conditions, and (2) a lack of evaluation in more complex noise scenarios, such as instance-dependent noise (IDN), class imbalance [53], or out-of-distribution (OOD) noise [54]. In future work, we aim to address the issue of high variance in OF1 and CF1 scores and to validate the model's robustness under a broader range of noise conditions and real-world multi-label noisy datasets. Additionally, part of CAD's overall performance gain can be attributed to the use of a pre-trained encoder. Moving forward, we plan to integrate discriminative and generative paradigms, enabling information generated by the diffusion model to guide various stages of training.

