# OpenReview forum: "Noisy Multi-Label Learning through Co-Occurrence-Aware Diffusion"
_NeurIPS.cc/2025/Conference — NeurIPS 2025 poster_

### Official Review · Reviewer_kEsb · 2025-06-27

**Clarity:** 2
**Significance:** 2
**Originality:** 2
**Rating:** 5
**Confidence:** 3

**Summary:**

The paper proposes a diffusion-based method coupled with neighborhood-based co-occurrence to handle the multi-label learning problem. In particular, the proposed method is inspired by the previous method CARD designed on single noisy label learning by firstly estimating the noisy multi-label distribution through the K nearest neighbor approach. Secondly, the obtained distribution of noisy multi-label matrix follows the diffusion process, and then reconstructed in the backward diffusion process as done in DDPM. The empirical evaluation on both synthetic and real-world datasets show the remarkable performance of the proposed method compared to the existing ones.

**Questions:**

Please address the two questions raised in the weaknesses. In particular, please explicitly provide the distinctions between CARD and the proposed method.

**Ethical Concerns:**

["NO or VERY MINOR ethics concerns only"]

**Final Justification:**

The authors have addressed most of my concerns during the discussion period.

**Limitations:**

The paper briefly discusses the limitations of the proposed method, which are more in the empirical evaluation, such as instance-dependent noise or out-of-distribution noise. There is no explicit discussion on the limitations of the proposed method.

**Quality:**

2

**Strengths And Weaknesses:**

## Strengths:
The paper is motivated from the real world setting where noisy labels may be contaminated into the set of multi-labels provided. Hence, it is important to address the noisy multi-label learning problem. The paper also provides a good overview and literature review on the previous and existing studies in the field.

The explanation of the proposed method is easy to follow, especially the visualization. In addition, the paper also presents extensive empirical evaluation to demonstrate the capability of their proposed method.

## Weaknesses:
- The proposed method is limited in terms of novelty. As described in the paper, it is an adopted version of CARD in noisy single-label learning to apply to the multi-label learning. Due to this similarity, it reduces the originality of the proposed method.
- Another concern is that diffusion models, in general, are designed to estimate the distribution of data. In this case, the data is the noisy multi-labels. The nearest neighbor step in the forward pass already provides a rough estimation of that distribution. The backward pass is to provide a "finer grain" version of that distribution. However, it is considered as the clean label distribution that is "denoised" by the diffusion model.

## Minors:
- Inconsistent English: the majority of the paper follows the American English, but some are in British English. For example, line 24: "expert-labelled" ==> "expert-labeled"
- Eqs (4) and (5) have unnecessary line break for a single column paper.

---

> ### Author Rebuttal · Authors · 2025-07-27
>
> Thanks for your valuable suggestions. Below, we provide point-by-point responses to the weaknesses and questions you raised, hoping to address your concerns regarding our work.
>
> **W1&Q1**: The proposed method is limited in terms of novelty. As described in the paper, it is an adopted version of CARD in noisy single-label learning to apply to the multi-label learning. Due to this similarity, it reduces the originality of the proposed method.
>
> **A1**: Thanks for your comment. Compared with CARD, our work **focuses on analyzing and improving practical performance for the specific problem**. Based on the characteristics of the noisy multi-label learning (NML), we conduct targeted theoretical analysis and method design, including: (i) **introducing feature-conditional strategies for noise-robustness and providing theoretical applicability guarantees**, and (ii) **developing diffusion-based embedding and coupling of label co-occurrence constraints**.
>
> **First**, the original feature conditioning mechanism in CARD relies on a supervised classifier $f_\phi$ to define the forward noisy distribution as $q(y_T|x) \sim {N}(f_\phi(x),I)$. However, this mechanism becomes invalid in the presence of noisy labels, as only noisy distributions are available during training, which can lead to a biased $f_\phi$ early on. In our work, **we redesign the feature conditioning strategy to be robust to label noise with theoretical applicability guarantees**, allowing it to be directly injected into the diffusion process. This improves the model's ability to capture the structured correlation between features and multiple labels.
>
> **Second**, prior work (_Chen J et al. Label-retrieval-augmented diffusion models for learning from noisy labels. NeurIPS, 2023, 36: 66499-66517_) has not addressed label co-occurrence relationships in noisy multi-label settings. To remedy this, **we incorporate label co-occurrence constraints** into the reverse diffusion process, which helps the model identify and avoid generating implausible label combinations (e.g., _Elephant_ and _Boat_), thereby **improving both the plausibility and robustness of the generation path**.
>
> **Finally**, we conduct extensive experiments on both synthetic and real-world noisy datasets, demonstrating that our method significantly outperforms existing SOTA methods across multiple evaluation metrics. These results validate the effectiveness of our proposed improvements. In summary, while our approach is inspired by the foundational ideas in CARD, **the modeling and optimization mechanisms we proposed are clearly novel and practically valuable for the NML setting**, and not a simple application or replication of prior work.
>
> **W2**: Another concern is that diffusion models, in general, are designed to estimate the distribution of data. In this case, the data is the noisy multi-labels. The nearest neighbor step in the forward pass already provides a rough estimation of that distribution. The backward pass is to provide a "finer grain" version of that distribution. However, it is considered as the clean label distribution that is "denoised" by the diffusion model.
>
> **A2**: Thanks for your insightful observation. As you correctly pointed out, the label estimation in the forward process does provide a coarse-grained version of the label distribution. However, due to the limited representational capacity of the pre-trained encoder’s feature space, **this initial estimation often falls short of ideal performance**. As shown in Table 1 (to be added in the final version, the data originates from Tables 1 and 2 in main text and Appendix Figure F.1) below, even when using a powerful encoder like ViT, the estimated multi-label classification metrics, such as OF1 and CF1, are still far from satisfactory. For example, on the VOC 2007 dataset with 50% symmetric noise, the CF1 score barely reaches 55%. In contrast, **after applying the proposed co-occurrence-aware diffusion process, our method outperforms** advanced NML baselines like HLC (_Xia X et al. Holistic label correction for noisy multi-label classification. ICCV, 2023: 1483-1493._), even when using a coarse-grained feature space and initial label distribution from a weaker encoder like ResNet50. This improvement arises not merely from converting coarse to fine label distributions, **but more importantly from refining the feature space itself under the guidance of image features to achieve a better alignment with the target multi-label set**. In other words, if we draw an analogy between the label diffusion model and a classifier, the diffusion process can be viewed as learning a mapping between extracted features and multi-label targets. However, unlike traditional discriminative models that directly minimize classification loss, our diffusion framework optimizes the L2 distance ($||\epsilon-\epsilon_\theta(y_t,{x},t)||^2$) between the generated denoised label distributions and the noisy ground-truth labels across all time steps. This indirect training objective **encourages the model to learn a probabilistic generation path from features to labels, ultimately building a more robust generative mapping**. We will include Table 1 and the accompanying explanation in the final version of the paper to further clarify the mechanism and contribution of the diffusion paradigm in this context.
>
> Table 1: Comparison of label pre-estimation (𝑦̄) and full CAD model results (%) using different pre-trained encoders.
> |Method|Pre-trained encoder|VOC2007-sym50%|||VOC2007-asym50%|||VOC2012-sym50%|||VOC2012-asym50%|||
> |---|---|:---:|:---:|:---:|:---:|:---:|:---:|:---:|:---:|:---:|:---:|:---:|:---:|
> | | |mAP|OF1|CF1|mAP|OF1|CF1|mAP|OF1|CF1|mAP|OF1|CF1|
> |𝑦̄|ResNet50|63.65|47.42|44.53|45.36|31.62|37.88|65.88|39.69|35.31|59.56|32.62|31.61|
> |CAD|ResNet50|72.37|67.19|64.13|61.88|59.51|56.17|75.53|69.80|64.86|68.36|61.60|60.56|
> |𝑦̄|ViT-14/L|82.20|53.21|55.43|52.38|39.99|46.57|82.27|48.64|47.79|52.17|44.23|48.68|
> |CAD|ViT-14/L|84.61|72.41|69.33|69.37|64.51|62.74|88.31|75.22|71.24|76.63|66.78|67.49|
> |HLC(Baseline)|N/A|68.03|66.62|63.65|59.09|58.59|56.51|69.03|67.83|64.37|64.27|60.64|59.86|
>
> **M1**: Inconsistent English: the majority of the paper follows the American English, but some are in British English. For example, line 24: "expert-labelled" ==> "expert-labeled".
>
> **A3**: Thank you for pointing this out. We appreciate your attention to detail. We will carefully revise the final version to ensure consistent use of American English throughout, including corrections like changing “expert-labelled” to “expert-labeled” and reviewing similar cases across the paper.
>
> **M2**: Eqs (4) and (5) have unnecessary line break for a single column paper.
>
> **A4**: Thanks for your careful review. We agree that the line breaks in Eqs. (4) and (5) are unnecessary, especially for a single-column format, and may affect readability. In the final version, we will revise these equations to be presented in a single line for better clarity and layout consistency.

---

> > ### Comment · Reviewer_kEsb · 2025-08-04
> > **Request for further clarification**
> >
> > The explanation from authors for the two concerns is still unclear. Could the authors elaborate further the following points:
> > - The difference (beyond single vs multiple noisy labels) between the current paper and another method in noisy label learning label-retrieval-augmented diffusion models (LRAD) [31]. In my initial review, I mistook [31] with [45].
> > - In the rebuttal, the authors claim about theoretical guarantee. As I read the Appendices which provide detailed deriation, to me, they are formulation for diffusion models. I cannot see anything that guarantee that the obtained label distribution would be the clean label distribution. Could the authors help to clarify?
> > - For my initial second concern, diffusion models, in general, are to estimate data distribution. For example, in the current paper, diffusion models are used to estimate the noisy label distribution of multiple labels $\Pr(\hat{y} | x)$. The noise in diffusion process is not the same as the noise in label noise. The explanation in the rebuttal has gone around and not addressed this point.

---

> > > ### Author Response · Authors · 2025-08-05
> > > **Response to Reviewer kEsb (Part 1, 2/3)**
> > >
> > > Thanks for your comments. We will respond point by point to address your concerns.
> > >
> > > **R1**: The Label Retrieval Augmented Diffusion model (LRAD) [31] is mainly designed for single-label noisy learning (LNL) and is an improved version of CARD [45]. Our method, CAD, differs from LRAD in two main aspects:
> > >
> > > (i) The label retrieval enhancement in LRAD searches for any sample within the k-nearest neighbors and directly uses its label as the estimated label. However, in the noisy multi-label learning (NML) problem, neighborhood samples in the feature space often have diverse label combinations. This random sampling strategy in LRAD introduces larger estimation bias. In contrast, our CAD estimates proxy points through a neighborhood interval estimation approach, which significantly improves estimation performance compared to LRAD (see Appendix F and Figure F.1 for detailed comparisons and analysis).
> > >
> > > (ii) Apart from the label retrieval enhancement, LRAD does not include any additional mechanisms to handle noisy labels, whether for single-label or multi-label data. Its frozen neighborhood feature space is used only for neighbor search. Our CAD further utilizes the frozen neighborhood feature space to estimate a label co-occurrence matrix, which is then used to impose a co-occurrence constraint during the reverse process of the diffusion model. The forward estimation and reverse co-occurrence constraint in CAD together form an EM-like model optimization strategy. This helps the diffusion model avoid learning incorrect multi-label distributions and reduces the impact of label noise.
> > > In summary, although LRAD can be applied to multi-label noisy learning as a type of LNL method, it is not specifically designed for NML problems, which leads to suboptimal performance (see Tables 1, 2, and 3 in main text). Our method is designed with the NML setting in mind and explores how to better apply and optimize the diffusion paradigm for this task. We hope this response addresses your concerns.
> > >
> > > **R2**: First, we would like to clarify that the main modeling target of the diffusion model is the ground-truth labels, which can also be referred to as clean labels in noisy scenarios (We will unify these two terms as much as possible in the final version). The main purpose of the theoretical derivation in Appendix is to ensure that the diffusion model operates properly within the label diffusion framework. By analyzing and designing a reasonable way to introduce features, we adapt the model to the NML task, allowing the diffusion model to learn the generative mapping from features to ground-truth (clean) multi-label distributions. We elaborate further with the following two points:
> > >
> > > (i) The diffusion model cannot directly model the generation process of noisy labels. As we stated in our Rebuttal’s A2, its essential modeling objective is the same as that of a classification model—minimizing the expected discrepancy between the inferred labels and the ground-truth labels. By extending the optimization process of each sample to $T$ steps, the diffusion model is able to learn generative class information that differs from discriminative models. The diffusion starting point $y_0$ and the feature condition play a key role in guiding the diffusion model to learn the mapping from features to clean multi-labels (see Equations (6) and (7)). A biased $y_0$ or an incorrect way of introducing the feature condition may lead the model to learn a mismatched mapping from features to labels.
> > >
> > > (ii) The presence of noisy labels can affect both $y_0$ and the guidance from feature conditions. CARD [45] introduces a feature condition by training a classifier $f_ \phi$​ on the training set and incorporating it into the forward process. Clearly, this approach is not suitable in noisy settings—noisy training data will contaminate the feature condition early on, causing the forward diffusion process to be biased. Meanwhile, the label retrieval technique in LRAD [31] results in a large deviation of $y_0$​ from the ground-truth labels (see our response to R1, point (ii)), which similarly causes a biased learning objective for the diffusion model.
> > >
> > > We discard the potentially noise-sensitive feature introduction method using the redundant classifier $f_ \phi$​ in CARD. Instead, by analyzing the mechanism of multi-label diffusion, we propose embedding a learnable image encoder (such as ResNet) into the decoder layer of the label diffusion model to inject feature-to-label guidance. The label estimation technique in CAD alleviates the $y_0$​ estimation bias present in LRAD, further improving the diffusion model’s accuracy in modeling the distribution of ground-truth (clean) labels. Our theoretical derivation generally supports these two improvements. In the final version, we will refine the description of the theoretical guarantees and add explanatory text related to the derivations supporting these design improvements.

---

> > > > ### Author Response · Authors · 2025-08-05
> > > > **Response to Reviewer kEsb (Part 2, 3/3)**
> > > >
> > > > **R3**: Your understanding is correct—the noise in the diffusion process is not the same as label noise. Specifically, the optimization objective of the diffusion model is to minimize the L2 distance between the predicted noise distribution and the true noise distribution, i.e., $|| \epsilon - \epsilon_ \theta ||^2$. This noise is not related to label noise from image features; rather, it refers to Gaussian noise introduced to help the diffusion model learn generative knowledge.
> > > >
> > > > Second, in our current paper, the diffusion model does not directly estimate noisy multi-labels. We apologize if the original wording in the paper caused any misunderstanding. In classification tasks, the main goal of the label diffusion model is to estimate the ground-truth (clean) labels—this is also the case for our method. When applied to noisy multi-label problems, this modeling approach can be seen as learning on noisy multi-label data (from a dataset-level perspective), but not as explicitly estimating the noisy labels themselves. Therefore, in CAD, we first use a label estimation technique to recover a $y_0$​ that approximates the ground-truth multi-label distribution as closely as possible. Then, using the co-occurrence constraint and feature condition guidance, we lead the diffusion model to learn a generative mapping from features to the corresponding multi-labels. After training, the diffusion model is able to infer a clean (ground-truth) multi-label distribution for any given input pair of $(x,y_T \sim N(0,I))$ (see Appendix C and Algorithm C.1 for detailed inference process for the clean labels).
> > > >
> > > > Thank you again for your attention and thoughtful discussion. We hope this response addresses your concerns. We will revise and clarify any ambiguous statements in the final version of the paper.

---

### Official Review · Reviewer_FnCb · 2025-06-29

**Clarity:** 3
**Significance:** 2
**Originality:** 3
**Rating:** 5
**Confidence:** 4

**Summary:**

This paper proposes a Co-Occurrence-Aware Diffusion (CAD) model, which reformulates noisy multi-label learning (NML) from a generative perspective.  It treats features as conditions and multi-labels as diffusion targets, optimizing the diffusion model for multi-label learning with theoretical guarantees.  To mitigate the interference of noisy labels in the forward process, it guides generation using pseudo-clean labels reconstructed from the latent neighborhood space, replacing original point-wise estimates with neighborhood-based proxies.  In the reverse process, it incorporates label co-occurrence constraints to enhance the model’s awareness of incorrect generation directions, thereby promoting robust optimization.  Experiments demonstrate the effectiveness of the proposed method.  The main contributions of this paper are:

- It reframes NML as a robust label generation task based on diffusion models with theoretical deduction, and enhances the mapping between features and multi-labels through matrix-based label representation.
- It designs a pseudo-clean label reconstructor and a meta-label co-occurrence matrix estimator, leveraging pre-trained encoders to provide strong priors for diffusion model training.
- It integrates co-occurrence constraints into the diffusion modeling, proposing the Co-Occurrence-Aware Diffusion (CAD) model, which can robustly learn the generative mapping from features to true multi-labels.

**Questions:**

1.  Why is a diffusion model preferred over other generative frameworks on the NML problem?
2.  How does CAD perform when trained from scratch or with a weakly initialized encoder?

**Ethical Concerns:**

["NO or VERY MINOR ethics concerns only"]

**Final Justification:**

After reading the comments from all reviewers and the authors' responses, I believe that the authors have addressed my concerns. Therefore, I support the acceptance of this paper.

**Limitations:**

The authors could further discuss the potential societal impact of the proposed method.

**Quality:**

3

**Strengths And Weaknesses:**

**Strength**

- The problem studied in this paper is interesting and valuable.
- This paper is well written and in good sharp, which is easy to follow.
- Experimental results are promising and can validate the effectiveness of the proposed method.



**Weakness**

- For the motivation of this paper, in my opinion, it does not clearly explain why a diffusion model is preferred over other generative frameworks on the NML problem.  Therefore, the practical requirements for CAD are somewhat unclear. The manuscript lacks a detailed discussion of the practical requirements and benefits that CAD fulfills over alternative frameworks. As a result, readers are left without a clear understanding of why the diffusion paradigm was chosen, what specific challenges in modeling noisy label matrices it addresses, and how its properties translate into tangible performance or stability advantages in the multi‑label denoising context.
- The proposed CAD seems to rely heavily on a strong pre‑trained encoder.  However, the paper does not explore scenarios in which such powerful pre‑trained models are unavailable, such as domains with limited data or modalities without large-scale pre-training assets. There is no evaluation of CAD‘s performance when the encoder is trained from scratch, initialized with random weights, or borrowed from a weaker backbone architecture.  Without such analysis, it remains unclear whether the benefits of CAD stem primarily from the diffusion framework itself or are largely inherited from the strength of the underlying feature representations.

---

> ### Author Rebuttal · Authors · 2025-07-27
>
> Thanks for your valuable suggestions. Below, we provide point-by-point responses to the weaknesses and questions you raised, hoping to address your concerns regarding our work.
>
> **W1&Q1**: Why is a diffusion model preferred over other generative frameworks on the NML problem?
>
> **A1**: Thanks for pointing out the need to clarify our motivation for choosing diffusion models over other generative frameworks for the NML problem. In real-world multi-label annotation scenarios, inherent uncertainty is common due to complex image semantics, ambiguous category boundaries, and subjective differences among annotators. Consequently, incomplete or conflicting annotations often lead to label noise, which is difficult to be avoided. Therefore, it is more realistic to view the multi-label learning process as an uncertainty-aware generation task conditioned on image features. Instead of directly fitting noisy labels, we aim to learn a distributional mapping from the original features to the “most likely true label set,” thereby enhancing robustness to label noise. To this end, we adopt diffusion models as the label generation framework. On the one hand, prior work (_Han X et al.. CARD: Classification and Regression Diffusion Models. NeurIPS, 2022_) has compared the modeling ability of diffusion models against other generative approaches such as GANs (e.g., GCDS, _Xiao Z et al. Tackling the Generative Learning Trilemma with Denoising Diffusion GAN. ICLR, 2022_), showing that **diffusion models are more effective at capturing predictive uncertainty in classification tasks**. This makes them particularly suitable for modeling the noise and ambiguity in label generation. On the other hand, in text-to-image generation, diffusion models have demonstrated the ability to synthesize images that combine multiple target concepts given composite prompts. This suggests that diffusion models are **inherently capable of capturing multiple objectives**. This indicates that in NML tasks, diffusion models may be well-suited to handle input images with multiple target objects, as their inherent ability to capture multiple objectives allows them to generate appropriate combinations of labels, i.e., an ideal characteristic for multi-label classification. Taken together, these characteristics **make diffusion models a natural and powerful choice for addressing the challenges in NML**. We will incorporate a more detailed discussion of these motivation and comparative analysis in the final version of the paper.
>
> **W2&Q2**: How does CAD perform when trained from scratch or with a weakly initialized encoder?
>
> **A2**: We greatly appreciate your concern regarding the use of pre-trained models. Please allow us to address your question from two perspectives.
>
> **First**, it is important to emphasize that our **CAD model does not heavily depend on the pre-trained encoder**, we can train a feature encoder from scratch using unsupervised approaches like SimCLR (_Chen T et al. A simple framework for contrastive learning of visual representations. ICML, 2020: 1597-1607_). Table 1 (to be added in the final version) shows the intermediate neighborhood label estimation and final classification results of CAD when combined with different pre-trained encoders include scratched ResNet50. The ResNet50 trained via unsupervised feature learning on the target dataset surpasses the ImageNet-pretrained counterpart in both label pre-estimation accuracy and performance when integrated with CAD, primarily because it effectively constitutes a form of fine-tuning on the target data distribution. On the other hand, from your perspective, ResNet50 can be viewed as a weakly initialized encoder compared to ViT-L/14, and the neighborhood label estimation alone based on ResNet50 performs worse than the baseline NML method across key metrics like OF1 and CF1. Even with such coarse estimations, CAD is capable of refining the generative feature-label mapping and consistently achieving performance improvements beyond the label estimation stage, outperforming the baseline NML method. This clearly highlights **the indispensable role of the diffusion architecture in enhancing performance beyond what weak feature spaces can offer**.
>
> Table 1: Comparison of label pre-estimation (𝑦̄) and full CAD model results (%) using different pre-trained encoders.
>
> |Method|Pre-trained encoder|VOC2007-sym50%|||VOC2007-asym50%|||VOC2012-sym50%|||VOC2012-asym50%|||
> |---|---|:---:|:---:|:---:|:---:|:---:|:---:|:---:|:---:|:---:|:---:|:---:|:---:|
> | | |mAP|OF1|CF1|mAP|OF1|CF1|mAP|OF1|CF1|mAP|OF1|CF1|
> |𝑦̄|ResNet50 (scratch)|65.49|48.77|45.83|46.78|32.59|38.92|67.62|40.84|36.37|61.18|33.53|32.55|
> |CAD|ResNet50 (scratch)|74.65|69.31|66.19|63.85|61.45|57.93|77.85|71.87|66.83|70.47|63.46|62.33|
> |𝑦̄|ResNet50 (pre-trained)|63.65|47.42|44.53|45.36|31.62|37.88|65.88|39.69|35.31|59.56|32.62|31.61|
> |CAD|ResNet50 (pre-trained)|72.37|67.19|64.13|61.88|59.51|56.17|75.53|69.80|64.86|68.36|61.60|60.56|
> |𝑦̄|ViT-14/L|82.20|53.21|55.43|52.38|39.99|46.57|82.27|48.64|47.79|52.17|44.23|48.68|
> |CAD|ViT-14/L|84.61|72.41|69.33|69.37|64.51|62.74|88.31|75.22|71.24|76.63|66.78|67.49|
> |HLC(Baseline)|N/A|68.03|66.62|63.65|59.09|58.59|56.51|69.03|67.83|64.37|64.27|60.64|59.86|
>
> **Second**, recent studies (_Zhu Z et al. Detecting corrupted labels without training a model to predict. ICML. 2022: 27412-27427._; _Ko J, Ahn S, Yun S Y. Efficient utilization of pre-trained model for learning with noisy labels. ICLR 2023_ and _Feng C, Tzimiropoulos G, Patras I. Clipcleaner: Cleaning noisy labels with clip. ACM MM, 2024: 876-885_) increasingly explore how to leverage pre-trained models with noise-free knowledge to handle label noise without additional training. These works, aligned with our view, suggest that **utilizing simple, accessible, and easily integrated pre-trained encoders is a practical and reasonable** strategy. The weights used for both ResNet50 and ViT-L/14 are publicly available and pre-trained on ImageNet—commonly adopted in most classification settings. For instance, using ResNet simply requires setting $pretrained=True$. While it is possible to train a feature encoder from scratch using unsupervised approaches like SimCLR (_Chen T et al. A simple framework for contrastive learning of visual representations. ICML, 2020: 1597-1607_), we believe that, **when high-quality, readily available pre-trained encoders exist, it is both reasonable and efficient to use them**. To ensure fair comparison, we selected LRAD and LSNPC as our main baselines, both of which also use pre-trained encoders. Table 2 (The data originates from Appendix Table E.1) shows that under the same latent feature space conditions, **CAD consistently makes better use of latent knowledge and refines it into more effective multi-label mappings**. We will include Table 1 in the final version to illustrate the model’s non-reliance on specific pre-trained encoders and will add a more detailed discussion on this issue.
>
> Table 2: Results (%) on VOC 2012 and MS-COCO with 30% symmetric noise, using various methods with different pre-trained encoders.
> |Method|Pre-trained|VOC 2012||MS-COCO||
> |---|---|:---:|:---:|:---:|:---:|
> |||mAP|OF1|mAP|OF1|
> |LRAD|ResNet50|71.69|61.60|59.55|56.06|
> |LSNPC|ResNet50|75.07|66.58|62.40|60.32|
> |CAD|ResNet50|77.02|72.27|65.78|63.70|
> |LRAD|ViT-L/14|81.15|71.31|69.91|65.76|
> |LSNPC|ViT-L/14|82.16|72.21|68.85|64.27|
> |CAD|ViT-L/14|88.99|77.33|73.93|70.39|

---

> > ### Comment · Reviewer_FnCb · 2025-08-05
> >
> > Thank you for the detailed responses. The rebuttal addresses my concerns. It is encouraged for the authors to further refine the paper according to the reviewers' comments.

---

> > > ### Author Response · Authors · 2025-08-05
> > > **Response to Reviewer FnCb**
> > >
> > > We are glad to have addressed your concerns. In the final version, we will follow your suggestions and include the motivation for using diffusion models, as well as experiments and analysis related to the initialization of pretrained models. Thank you again for your valuable feedback.

---

### Official Review · Reviewer_UB3E · 2025-07-03

**Clarity:** 3
**Significance:** 2
**Originality:** 3
**Rating:** 4
**Confidence:** 4

**Summary:**

The paper proposes to learn the posterior mapping from features to labels in the noisy multi-label problem using the diffusion model. The noisy issue is constrained by using pseudo-clean labels in the generation process. The pseudo-clean labels are obtained from the latent neighborhood space, replacing the point-wise estimates in traditional methods. The proposed method is evaluated on both synthetic and real-world noisy datasets.

**Questions:**

1. What happens if the label correlation is less associated with the latent space similarity? What happens when the domain knowledge is very different from pre-trained knowledge?

2. What are the exact theoretical guarantees?

3. What is the performance on rare labels?

**Ethical Concerns:**

["NO or VERY MINOR ethics concerns only"]

**Final Justification:**

The paper is technically sound and makes a decent contribution. The rebuttal is adequate but some concerns still remain. Thus I will keep my rating.

**Limitations:**

Yes

**Quality:**

3

**Strengths And Weaknesses:**

Strengths:

1. The paper is well written and most information is clearly presented.

2. The idea of using the diffusion model to address the noisy multi-label learning problems is novel and interesting.

3. The experiments show that the proposed method has an advantage compared to existing noisy multi-label learning methods.

Weaknesses:

1. The proposed method seems dependent on the label correlation being associated with the latent space similarity. However, in some tasks and domains, it might not be applicable. In these cases, the diffusion model generation might not meet the requirement for pseudo-clean labels. There might also be issues when there are both noisy labels and out-of-distribution data samples. These are not well discussed in the paper.

2. The paper mentions theoretical guarantees, but it is unclear what the guarantees are referring to.

3. The paper does not discuss other possible ways of mitigating the noise issue using the latent neighborhood space, such as other automatic correction models.

4. The paper does not discuss the possibility of damaging the rare labels, which may occur simultaneously with other unexpected labels.

---

> ### Author Rebuttal · Authors · 2025-07-27
>
> Thanks for your valuable suggestions. Below, we provide point-by-point responses to the weaknesses and questions you raised, hoping to address your concerns regarding our work.
>
> **W1&Q1**:  What happens if the label correlation is less associated with the latent space similarity? What happens when the domain knowledge is very different from pre-trained knowledge?
>
> **A1**: Thanks for your insightful observation regarding the dependence of the proposed CAD model on the strength of association between label correlations and the pretrained latent feature space.
>
> **First**, Table 1 (originates from Appendix Table E.1) shows that the contribution of different pre-trained models to the latent feature space used in CAD. Both models are pre-trained on ImageNet. From your perspective, it is reasonable to interpret that **ResNet50 exhibits a weaker association between label correlations and its latent space compared to ViT**. This is primarily due to the relatively limited representational capacity of ResNet50’s latent space. Consequently, the performance of **CAD combined with ResNet50 is weaker than when combined with ViT** or other pre-trained models with stronger representational power (e.g., ADDGCN, HLC). However, compared to other methods that also leverage pre-trained models to extract latent space knowledge (e.g., LSNPC, LRAD), **our method consistently makes better use of this coarse-grained feature space**. Through a diffusion paradigm, CAD refines these features into a generative mapping for multi-label classification, resulting in superior performance despite relying on the same latent knowledge. Furthermore, even when CAD is combined with a weaker feature space such as that of ResNet50, it still shows significant advantages over methods that do not utilize pre-trained models at all. **Efficiently leveraging simple, offline, and easily accessible pre-trained models to obtain usable latent feature spaces  is one of the strengths of our method** and aligns with strategies recognized as effective in the current studies (_Ko J et al. Efficient utilization of pre-trained model for learning with noisy labels. ICLR, 2023_ and  _Zhu Z et al. Detecting corrupted labels without training a model to predict. ICML, 2022: 27412-27427_).
>
> Table 1: Results (%) on VOC 2012 and MS-COCO with 30% symmetric noise, using various methods with different pre-trained encoders.
> |Method|Pre-trained|VOC 2012||MS-COCO||
> |---|---|:---:|:---:|:---:|:---:|
> |||mAP|OF1|mAP|OF1|
> |LRAD|ResNet50|71.69|61.60|59.55|56.06|
> |LSNPC|ResNet50|75.07|66.58|62.40|60.32|
> |CAD|ResNet50|77.02|72.27|65.78|63.70|
> |LRAD|ViT-L/14|81.15|71.31|69.91|65.76|
> |LSNPC|ViT-L/14|82.16|72.21|68.85|64.27|
> |CAD|ViT-L/14|88.99|77.33|73.93|70.39|
>
> **Second**, regarding the issue of out-of-distribution noise and potential misalignment between the pre-trained model and domain-specific knowledge, **one possible solution is to apply unsupervised fine-tuning methods**, such as SimCLR, on the pre-trained model before freezing it to obtain the latent feature space. Based on our empirical experiments, combining CAD with the ViT model allows us to effectively utilize a wide range of image datasets, including those involving fine-grained classification tasks. Prior research (_Wei T et al. Vision-language models are strong noisy label detectors. NeurIPS, 2024, 37: 58154-58173_) has also demonstrated the effectiveness of fine-tuning ViT encoders to address noise detection under different data distributions. We will include a discussion of this limitation, along with potential directions for future work, in the revised "Limitations" section.
>
> **W2&Q2**: What are the exact theoretical guarantees?
>
> **A2**: Our theoretical contribution primarily lies in the development of a label diffusion paradigm designed to simultaneously address multi-label classification and label noise. While CARD has introduced a label diffusion model for single-label classification, it does not provide a solution for either multi-label settings or noisy labels. For example, CARD’s feature conditioning mechanism relies on training an auxiliary classifier in advance on the dataset to guide the forward diffusion process with injected noise. However, in noisy label environments, this reliance often infeasible. On the other hand, LRAD addresses label noise via label estimation but does not consider label co-occurrence relationships inherent in multi-label tasks, limiting its effectiveness for noisy multi-label scenarios. In contrast, **our work theoretically derives both unconditional and conditional multi-label diffusion processes and proposes a straightforward, noise-robust method for feature conditioning**. This leads to a diffusion network architecture that directly aligns with the proposed theoretical framework. To reduce potential ambiguity, we will revise the final version of the paper by replacing “theoretical guarantees” with “**theoretical applicability guarantees**” in both the abstract and main text. This will clarify that our contribution lies in demonstrating the applicability of diffusion theory to noisy multi-label learning tasks.
>
> **W3**: The paper does not discuss other possible ways of mitigating the noise issue using the latent neighborhood space, such as other automatic correction models.
>
> **A3**: Unlike most methods that iteratively correct labels during model training, our CAD model performs a single-step pre-correction process through the latent feature space, without requiring any additional training or fine-tuning. To further validate this property, we introduced two additional label correction modules that also operate in a training-free and one-step manner: SimiFeat (_Zhu Z et al. Detecting corrupted labels without training a model to predict. ICML, 2022: 27412-27427_) and CLIPCleaner (_Feng C et al. Clipcleaner: Cleaning noisy labels with clip. ACM MM, 2024: 876-885._). SimiFeat leverages a frozen feature extractor and label consistency among similar samples, using a Bayesian thresholding strategy to detect and correct noisy labels. CLIPCleaner aligns image and text embeddings using CLIP’s encoders to perform semantic-based label correction. A preliminary comparison of these two modules with our proposed estimation method (𝑦̄) is presented in Table 2 below. SimiFeat demonstrates superior correction performance, while CLIPCleaner appears to be less effective, possibly due to the limited accuracy of prompt-based semantics in multi-label contexts. When integrated with CAD, we observe that **more accurate pre-correction leads to further performance improvements** of the proposed diffusion paradigm. Our contribution lies in using the diffusion model to refine image features into a generative multi-label mapping based on pre-corrected labels. The integration with different correction modules also demonstrates **the generality and extensibility of the proposed CAD**. We will include a more comprehensive comparison of automatic correction models combined with CAD in the final version, and expand the discussion in the related work section accordingly.
>
> Table 2: Performance (%) comparison of different automatic correction models and their integration with CAD on VOC 2012.
> |Method|Sym. 50%|||Asym. 50%|||
> |---|:---:|:---:|:---:|:---:|:---:|:---:|
> ||mAP|OF1|CF1|mAP|OF1|CF1|
> |𝑦̄|82.27|48.64|47.79|52.17|44.23|48.68|
> |SimiFeat|85.11|50.25|49.22|53.79|45.61|50.17|
> |CLIPCleaner|80.39|47.63|46.62|50.83|43.01|47.63|
> |𝑦̄+CAD|88.31|75.22|71.24|76.63|66.78|67.49|
> |SimiFeat+CAD|89.16|76.13|72.85|77.31|67.82|68.41|
> |CLIPCleaner+CAD|87.03|74.23|70.04|75.57|65.24|66.38|
>
> **W4&Q3**: The paper does not discuss the possibility of damaging the rare labels, which may occur simultaneously with other unexpected labels. What is the performance on rare labels?
>
> **A4**: In noisy multi-label settings, rare labels may be mistakenly treated as label noise, which will negatively impact the model’s ability to fit them. In traditional weak supervision scenarios with extreme class imbalance, previous studies have proposed various semi-supervised and active learning approaches to address the challenge of rare label modeling. For example, graph-based label propagation methods (_Pimplikar R et al. Learning to propagate rare labels. ACM CIKM, 2014: 201-210_)  leverage difference-of-convex optimization and k-NN subgraph sampling to improve robustness to extreme imbalance in large-scale graph-structured data. Alternatively, pseudo-labeling and representation-aware active learning techniques can be used to train robust models that mitigate the impact of rare labels (_Mullapudi R T et al. Learning rare category classifiers on a tight labeling budget. ICCV, 2021: 8423-8432_). Although our method is not specifically designed to address the rare label problem, we think the proposed **CAD can partially alleviate this issue by expanding neighborhood search, or strengthening co-occurrence constraints via advanced matrix estimation techniques**. The datasets used in our experiments are popular for evaluating NML methods, and they **usually do not exhibit extreme imbalance**. For instance, in the VOC dataset (excluding the most basic class person), the lowest label frequency (cow, 273 instances) and the highest (chair, 1168 instances) yield a ratio of roughly 1:4. Even if common noise is added to these extreme labels, it is still less likely for rare label situations to occur. In view of this, the proposed **CAD does not include strategies specifically designed to address the rare label problem**.  Alternatively, We will briefly discuss this limitation in the final version of the paper. In future work, we plan to explore both simulated and real-world imbalanced multi-label datasets to better capture rare label scenarios and to develop dedicated solutions that can be integrated with the diffusion paradigm for enhanced performance.

---

> > ### Comment · Reviewer_UB3E · 2025-08-05
> >
> > Thank you for the response and additional results. I appreciate the answers on label correlation impact and comparison of automatic correction models. I remain reserved about the theoretical guarantees and rare label questions. Thus, I will most likely maintain the current rating for the paper.

---

> > > ### Author Response · Authors · 2025-08-06
> > > **Response to Reviewer UB3E**
> > >
> > > Thank you for your continued feedback. We understand and respect your reservations. If possible, we would like to briefly clarify two points regarding the theoretical guarantees and rare label issues:
> > >
> > > **(i) Theoretical guarantees**: The main purpose of the theoretical derivation in Appendix A—C is to ensure that the diffusion model operates properly within the label diffusion framework. By analyzing and designing a reasonable way to introduce features, we adapt the model to the NML task, allowing the diffusion model to learn the generative mapping from features to ground-truth multi-label distributions.
> > >
> > > **(ii) Rare label issue**: Your concern about rare labels is insightful. This is indeed a challenging aspect in classification, and most existing noisy multi-label learning (NML) methods do not explicitly address it. Rare labels may be one of the key factors limiting model performance. Your comment brings valuable attention to this issue and opens up new directions for future research in NML. Our CAD does not include specific handling for rare labels either, but in the final version, we plan to add small-scale experiments to highlight this limitation and point to it as a promising direction for future work.
> > >
> > > Thank you again for your thoughtful review, valuable suggestions, and ongoing discussion. We truly appreciate your time and effort in evaluating our work. If you have any further suggestions or concerns, particularly regarding the theoretical guarantees and the rare label issue, we would be more than happy to discuss them with you to improve the paper, either in the final version or in future research.

---

### Official Review · Reviewer_v11J · 2025-07-04

**Clarity:** 3
**Significance:** 3
**Originality:** 3
**Rating:** 4
**Confidence:** 2

**Summary:**

The paper reframes NML as a robust label generation task based on diffusion models with theoretical deduction, and enhances the mapping between features and multi-labels through matrix-based label representation. The paper designs a pseudo-clean label reconstructor and a meta-label co-occurrence matrix estimator, leveraging pre-trained encoders to provide strong priors for diffusion model training. Besides, the paper integrates co-occurrence constraints into the diffusion modeling, proposing the Co-Occurrence-Aware Diffusion (CAD) model, which can robustly learn the generative mapping from features to true multi-labels.

**Questions:**

(1) It would be great if the paper can provide some analysis related to the computational costs of the developed approach and the baselines.

(2) It would be great to provide more case study results to illustrate how the developed key components work in practice. Some intermediate results may also be presented.

(3) Some discussed previous works (e.g., [20]) provide theoretical results. Might it be feasible to provide some theoretical results similarly in this paper? Some discussions would be very insightful.

**Ethical Concerns:**

["NO or VERY MINOR ethics concerns only"]

**Final Justification:**

The majority of my questions were answered during the discussion, so I will keep my positive rating.

**Limitations:**

yes

**Quality:**

3

**Strengths And Weaknesses:**

Strengths

(1) The paper is well motivated and is related to an important task in practice.

(2) The paper formulation is novel and the developed approach has good intuitions.

(3) The experiments are comprehensive and the developed approach achieves strong performances.

Weaknesses

(1) It would be great if the paper can provide some analysis related to the computational costs of the developed approach and the baselines.

(2) It would be great to provide more case study results to illustrate how the developed key components work in practice. Some intermediate results may also be presented.

---

> ### Author Rebuttal · Authors · 2025-07-26
>
> Thank you for your insightful suggestions. Below, we provide point-by-point responses to the weaknesses and questions you raised, hoping to address your concerns regarding our work.
>
> **W1&Q1**: It would be great if the paper can provide some analysis related to the computational costs of the developed approach and the baselines.
>
> **A1**: We have provided a more detailed analysis of the training and inference time costs for our proposed CAD method and the baseline methods in Tables 1 and 2 below (The data originates from Appendix Figure H.1 and Table H.1). Table 1 shows that CAD achieves **better training efficiency** compared to mainstream approaches due to the lightweight diffusion architecture. Table 2 demonstrates that, with the accelerated DDIM sampling strategy, the time cost of generative multi-label inference is on the same order of magnitude as that of discriminative prediction, indicating that CAD is **suitable for large-scale data scenarios**. We will incorporate this part of the Appendix into the main text in the final version.
>
> Table 1:  Time cost (sec) of different method for training one epoch on one NVIDIA A800 GPU.
> |Dataset|LSNPC|HLC|ASL|CAD(Ours)|LRAD|
> |---|:---:|:---:|:---:|:---:|:---:|
> |VOC2007|25|21|13|9|8|
> |VOC2012|40|31|23|20|15|
> |MS-COCO|462|393|379|357|330|
>
> Table 2: Test time cost (sec) for CAD and discriminative (Disc.) models on different test sets.
> |Dataset|CAD|Disc.|
> |---|:---:|:---:|
> |VOC(4.9kimgs)|8|5|
> |COCO(40kimgs)|173|102|
>
> **W2&Q2**: It would be great to provide more case study results to illustrate how the developed key components work in practice. Some intermediate results may also be presented.
>
> **A2**: In our CAD architecture, the neighborhood label estimation in the forward process serves as an initial coarse label guess produced by a non-diffusion model. The subsequent diffusion training process refines this estimate by incorporating feature and co-occurrence constraints. As shown in Table 3 below (The data originates from Table 1 and Table 2 in main text and Appendix Figure F.1), which includes results from our ablation study, **the diffusion training consistently improves multi-label classification performance based on the initial label estimates**. This demonstrates that our training framework enables the diffusion model to effectively learn a robust generative mapping from features to multi-label outputs.
>
> Table 3: Comparison of label pre-estimation (𝑦̄) and full CAD model results (%).
> |Method|VOC2007-sym50%|||VOC2007-asym50%|||VOC2012-sym50%|||VOC2012-asym50%|||
> |---|:---:|:---:|:---:|:---:|:---:|:---:|:---:|:---:|:---:|:---:|:---:|:---:|
> | |mAP|OF1|CF1|mAP|OF1|CF1|mAP|OF1|CF1|mAP|OF1|CF1|
> |𝑦̄|82.20|53.21|55.43|52.38|39.99|46.57|82.27|48.64|47.79|52.17|44.23|48.68|
> |CAD|84.61|72.41|69.33|69.37|64.51|62.74|88.31|75.22|71.24|76.63|66.78|67.49|
>
> **Q3**: Some discussed previous works (e.g., [20]) provide theoretical results. Might it be feasible to provide some theoretical results similarly in this paper? Some discussions would be very insightful.
>
> **A3**: Thanks for your constructive suggestion regarding the theoretical aspects of our work. Indeed, [20] provides valuable theoretical insights on using label correlations for estimating the noise transition matrix (LCT). In principle, it is feasible to incorporate such estimation techniques such as T-estimator (_Liu T et al. Classification with noisy labels by importance reweighting. TPAMI, 2015, 38(3): 447-461_) and Dual T-estimator (_Yao Y et al. Dual t: Reducing estimation error for transition matrix in label-noise learning. NeurIPS, 2020, 33: 7260-7271_) into our CAD architecture, since our co-occurrence matrix is also estimated by observing label co-occurrence probabilities within a clean subset. However, these methods typically **require additional network components and warm-up training**. For example, [20] estimates the transition matrix for each class $i$ by considering all other classes $k$ (where $k ≠ i$), which will increase computational overhead.
> That said, we agree that employing more accurate co-occurrence estimation methods could potentially enhance the performance of our CAD framework. We conducted a small-scale experiment by replacing our co-occurrence constraint with reweighting strategies based on T-estimator and the method proposed in [20]. As shown in Table 4, all these alternatives led to **further performance improvements** for our method. We will include these experimental results in the final version and incorporate theoretical discussions from the matrix estimation literature to improve the interpretability of our co-occurrence modeling.
>
> Table 4: Comparison of the combined performance (%) of different matrix estimation reweighting methods with CAD under a 30% symmetric noise environment.
> |Method|VOC2012||COCO||
> |---|:---:|:---:|:---:|:---:|
> ||mAP|OF1|mAP|OF1|
> |CAD|88.99|77.33|73.93|70.39|
> |CAD+Reweight-T|**89.12**|78.02|74.13|70.52|
> |CAD+Reweight-LCT|89.07|**78.81**|**74.28**|**71.10**|

---

> > ### Comment · Reviewer_v11J · 2025-08-06
> >
> > Thank you very much for the additional responses and details. It would be great to include them in the updated version. I will keep the positive rating.

---

> > > ### Author Response · Authors · 2025-08-06
> > > **Response to Reviewer v11J**
> > >
> > > Thank you very much for your review, thoughtful suggestions, and dedicated effort throughout the entire review process. We sincerely appreciate your positive recognition of our work and will include the additional experiments and analysis from the Rebuttal in the final version.

---

### Note · Authors · 2025-08-12

Dear Area Chair and Reviewers,

We sincerely thank the Area Chair and reviewers for their time and effort in reviewing our submission.

In our manuscript, we propose a **generative perspective** to address the noisy multi-label learning (NML) problem. Our method constructs a comprehensive noisy multi-label diffusion architecture named **CAD** and provides theoretical derivations of its applicability. We also use neighborhood proxy estimation and **label co-occurrence constraints** to guide the multi-label diffusion training process, ensuring the model **robustly learns the generative mapping from features to multi-labels** on noisy datasets. Extensive experiments on both simulated and real-world noisy multi-label datasets demonstrate the effectiveness of our method and its superiority over various baseline approaches.

We greatly appreciate the positive feedback from the reviewers, who highlighted several strengths of our work. The NML problem addressed in this paper is considered **important and valuable** (Reviewer v11J, Reviewer kEsb). The idea of using a diffusion model to solve the NML problem was recognized as **novel and interesting, with good intuition** (Reviewer v11J, Reviewer UB3E, Reviewer FnCb). The explanation and formulation of the proposed method were noted as **clear, intuitive, and easy to follow** (All Reviewers). The experimental design and validation results were recognized as **comprehensive and promising** (All Reviewers).

During the rebuttal stage, we provided additional experiments for comparison, along with corresponding explanations and discussions.  Reviewer kEsb raised further questions such as **the differences between CAD and the previous LRAD**. In our discussion, we addressed these concerns by elaborating on aspects such as **the label pre-estimation and co-occurrence constraints designed for the NML problem**. We believe we have resolved most of the reviewers' concerns (for more details, please refer to our rebuttal and discussion).

We believe our CAD not only offers a novel generative solution to NML but also explores the application of diffusion models in image classification tasks. We again sincerely appreciate the constructive discussion and thank the reviewers and Area Chair for their dedication and valuable feedback, which have greatly improved our work. We commit to incorporating all improvements suggested during the rebuttal into the final version of the paper.

Best regards,

The Authors

---

### Decision · Program_Chairs · 2025-09-17

**Decision:**

Accept (poster)

**Comment:**

This paper reframes noisy multi-label learning through a co-occurrence-aware diffusion model (CAD) that conditions on image features.

The reviewers found the motivation clear and the core idea novel; two recommended acceptance and two were borderline-accept. The rebuttal and discussion significantly strengthened the paper by addressing key concerns:
(i) The empirical evaluation is comprehensive, with strong ablations, including tests on encoder sensitivity, where CAD shows improvements even with weaker or randomly initialized encoders.
(ii) Efficiency is well-quantified, with both training and inference times competitive compared to baselines.
Remaining concerns, such as theoretical guarantees and the handling of rare labels, have been acknowledged through clarifications and concrete plans for the camera-ready version.
Overall, the use of a diffusion model to tackle noisy multi-label learning is both novel and promising.